# Telecom-wavelength quantum teleportation using frequency-converted photons from remote quantum dots

Tim Strobel [1] ✉, Michal Vyvlecka [1], Ilenia Neureuther[1], Tobias Bauer [2], Marlon Schäfer [2], Stefan Kazmaier [1], Nand Lal Sharma[3], Raphael Joos [1], Jonas H. Weber [1], Cornelius Nawrath [1], Weijie Nie [3], Ghata Bhayani[3], Caspar Hopfmann [3], Christoph Becher [2], Peter Michler [1] & Simone Luca Portalupi [1]

A global quantum internet is based on scalable networks, which require reliable quantum hardware. Among them are quantum light sources providing deterministic, high-brightness, high-fidelity entangled photons and quantum memories with coherence times exceeding the millisecond range. Long-distance operation demands quantum light sources emitting at telecommunication wavelengths. A cornerstone for such networks is the demonstration of quantum teleportation. Here, we realize full-photonic quantum teleportation employing semiconductor quantum dots, which can fulfill all the aforementioned requirements. Two remote GaAs quantum dots, emitting in the near-infrared, are used: one as an entangled-photon pair source and the other as a single-photon source. During the experiment, the single photon is prepared in conjugate polarization states and interfaced with the biexciton emission of the entangled pair employing a polarization-selective Bell state measurement. This process teleports the respective polarization state onto the exciton emission of the entangled pair. The frequency mismatch between the triggered sources is erased using two polarization-preserving quantum frequency converters, enabling remote two-photon interference at telecommunication wavelengths, yielding a visibility of 30(1)%. A post-selected teleportation fidelity up to 0.721(33), significantly above the classical limit, demonstrates successful quantum teleportation between light from distinct sources. These results mark an important development for semiconductor-based quantum light sources.

In recent years, significant efforts have been made towards realizing the ambitious vision of a global quantum internet[1,2] that would allow to securely connect distant nodes, and to interface with remote quantum computers[3] or deployed quantum sensors. Fundamental for such a realization are quantum memories, to store and actively retrieve quantum information, and sources of quantum light to provide interconnection among different nodes. Several platforms are currently under investigation for their role in future quantum communication[4]:

[1]Institut für Halbleiteroptik und Funktionelle Grenzflächen, Center for Integrated Quantum Science and Technology (IQST) and SCoPE, University of Stuttgart, Stuttgart, Germany. [2]Fachrichtung Physik, Universität des Saarlandes, Saarbrücken, Germany. [3]Institute for Integrative Nanosciences, Leibniz IFW Dresden, Dresden, Germany. ✉e-mail: t.strobel@ihfg.uni-stuttgart.de

atoms and ions[5–8], defect centers in diamond[9–13], parametric processes[14–16], and semiconductor quantum dots (QDs)[17,18]. Because of recent milestone achievements demonstrating a tremendous improvement in spin coherence[19,20], epitaxial QDs showed great promise to function as quantum memory in future quantum networks. This is particularly appealing since they can effectively be interfaced with light generated by other QDs, which are known as efficient sources of single[21–28] and entangled-photons[29–36]. Recent studies also showed all-photonic schemes with indistinguishable photons from QDs in cluster states as memory-free alternatives[37,38].

A key resource in quantum communication is quantum teleportation[39–41], ideally realized with photons generated by remote sources of quantum light. Earlier studies with single QD emitters demonstrated their potential in teleportation experiments at near-infrared (NIR) wavelengths[42–45]. For successful implementation, photons capable of quantum interference and a high degree of entanglement are required. Furthermore, on-demand sources would be highly beneficial to upscale the network complexity, particularly when the generation process of single and entangled photons is deterministic and not probabilistic[46,47]. Moreover, the ability to tune emitter wavelengths to a common target wavelength is essential for ensuring photon indistinguishability between distant sources. Two-photon interference with light generated by distinct QD sources emitting in the NIR regime has been investigated[48–54] with recently reported high values of 93.0(8)% interference visibility[55]. In addition, if long-distance propagation needs to be achieved, employing standard optical silica fibers for connecting distant nodes is highly beneficial. Indeed, silica fibers already represent the backbone of global telecommunication infrastructure, where light at telecommunication wavelengths experiences minimal propagation losses and limited photon wavepacket dispersion. Such behavior is even more crucial for quantum light. While low loss would allow for a reduction in the number of required repeater stations, low dispersion would ensure high interference visibility for photons over channels with different lengths. These advantages render quantum light at telecommunication wavelengths particularly appealing for the future implementation of quantum communication. Despite the ongoing developments in realizing QD sources of quantum light operating at telecom wavelengths[36,56–60], state-of-the-art performances are still set by the QDs emitting at NIR wavelengths. For this reason, the use of frequency conversion was found to be an appealing approach to bridge this wavelength gap[61,62], and it was shown to be a powerful method for fine-tuning remote QD sources to the same wavelength, enabling two-photon interference[63,64]. Recently, quantum frequency converters designed to operate with QD light and capable of preserving the polarization state during conversion have been demonstrated[5,18,65].

Here, we make use of epitaxially grown droplet etching semiconductor GaAs QDs to implement a full-photonic quantum teleportation experiment employing two distinct semiconductor QD sources of triggered quantum light. The single-photon state generated by one source is teleported onto the second, non-interfering, entangled photon emitted by the second QD, by performing a Bell state measurement (BSM). Photon interference is achieved by using two independent polarization-preserving quantum frequency converters to eliminate the photons' wavelength mismatch, fully preserving the high degree of entanglement during conversion (see Supplementary Note 1G). Appealing for out-of-the-lab experiments, the converted photons exhibit telecommunication wavelengths, therefore enabling long-distance propagation along standard silica fibers. Furthermore, we study the photon teleportation dynamics via time-resolved measurements in detail. An average teleportation fidelity of up to 0.721(33), significantly above the classical limit, conclusively proves the success of teleportation in this full-photonic scheme. All the experimental results are confirmed by theoretical modeling, which explains each observed state's behavior in the teleportation process and enables a clear quantification of the envisioned possibilities in upcoming experiments.

## Results

### Experimental configuration

Figure 1a shows a general schematic of the experiment. Two remote QDs are used: QD1 functions as a single-photon source (SPS), emitting Photon 1. QD2 serves as an entangled pair source (EPS), emitting an entangled photon pair: Photon 2 and Photon 3. In both cases, the QDs are excited via pulsed two-photon excitation and generate photons via the biexciton-exciton cascade ($|XX\rangle \rightarrow |X\rangle \rightarrow |G\rangle$)[66–68]. The biexciton (XX) photons (Photon 1 and Photon 2) are sent to a BSM setup, after two distinct quantum frequency conversion (QFC) processes (see methods), while the exciton (X) emission of QD2 (Photon 3) is analyzed. The joint BSM projects Photon 1 and Photon 2 onto a maximally entangled Bell state, teleporting the polarization state of Photon 1 (named $|\xi\rangle_1$) onto Photon 3. The receiver reconstructs the polarization state of Photon 3 (named $|\xi\rangle_3$) conditioned on the BSM result.

Figure 1b depicts a detailed illustration of the experimental configuration. During the experimental procedure, a pulsed laser (304.8 MHz repetition rate corresponding to a 3.28 ns repetition period) coherently prepares a XX state in two epitaxially grown droplet etching GaAs QDs[29,30] remotely located in different cryostats. The excitation is followed by a cascaded emission of two polarization-entangled photons at NIR wavelengths (~780 nm). The polarization state $|\xi\rangle_1$ of Photon 1 (to be teleported) is prepared by sending it through a polarizing beamsplitter (PBS), followed by a half- (HWP) and a quarter-wave plate (QWP). Photon 2 and Photon 3 share a maximally entangled oscillating state $1/\sqrt{2}(|HH\rangle_{2,3} + e^{i\delta_2 t/\hbar}|VV\rangle_{2,3})$, where $|H\rangle$ ($|V\rangle$) represents horizontal (vertical) polarization, $\delta_2$ the fine-structure splitting (FSS) of the EPS, $t$ the time between XX and X emission, and $\hbar$ the reduced Planck constant. For slow FSS-induced oscillations (here $\delta_2 = 2.1(3)\,\mu$eV) relative to the emitter decay time, the latter state can be simplified to a maximally entangled Bell state $|\Phi^+\rangle_{2,3} = 1/\sqrt{2}(|HH\rangle_{2,3} + |VV\rangle_{2,3})$ (for more details see Supplementary Note 1G). The wavelengths of Photon 1 and Photon 2 at ~780 nm do not spectrally overlap, prohibiting interference. To enable two-photon interference required for a successful BSM, polarization-preserving QFC is employed (see Fig. 1b and methods)[65]. This process converts the XX photons (Photons 1 and 2) to a common telecommunication wavelength (1515 nm, see Fig. 2a), leaving their quantum state unaltered. Being at technologically relevant telecommunication wavelengths also opens the way for prospective long-distance teleportation experiments. After interference, the (three-photon) state can be written in the Bell basis:

$$|\Psi_{\text{tot}}\rangle = |\xi\rangle_1 \otimes |\Phi^+\rangle_{2,3} = \frac{1}{2}\left( |\Phi^+\rangle_{1,2}|\xi\rangle_3 + |\Phi^-\rangle_{1,2}\sigma_3|\xi\rangle_3 \right. \\ \left. + |\Psi^+\rangle_{1,2}\sigma_1|\xi\rangle_3 - |\Psi^-\rangle_{1,2}\sigma_1\sigma_3|\xi\rangle_3 \right), \quad (1)$$

with Pauli matrices $\sigma_1$ and $\sigma_3$ (for exemplary calculations see Supplementary Note 2B) and Bell states $|\Phi^\pm\rangle_{1,2} = 1/\sqrt{2}(|HH\rangle_{1,2} \pm |VV\rangle_{1,2})$ and $|\Psi^\pm\rangle_{1,2} = 1/\sqrt{2}(|HV\rangle_{1,2} \pm |VH\rangle_{1,2})$. Here, the BSM unit is implemented as a single-mode fiber beamsplitter (FBS) with a PBS at each output arm, followed by superconducting nanowire single-photon detectors (SNSPDs). In this polarization-selective BSM configuration the projections of Photon 1 and Photon 2 onto Bell states $|\Psi^+\rangle_{1,2}$ or $|\Psi^-\rangle_{1,2}$ can be identified, increasing the BSM efficiency considerably[45,69]. A successful BSM heralds the unitary transformation to be applied to Photon 3 to reconstruct the initially prepared quantum state of Photon 1 (see Supplementary Note 2B). To probe the teleported polarization state $|\xi\rangle_3$, Photon 3 is analyzed with a tomography unit consisting of a QWP, a HWP followed by a PBS, and two SNSPDs. In the following, the results for one projection onto $|\Psi^-\rangle_{1,2}$ are discussed

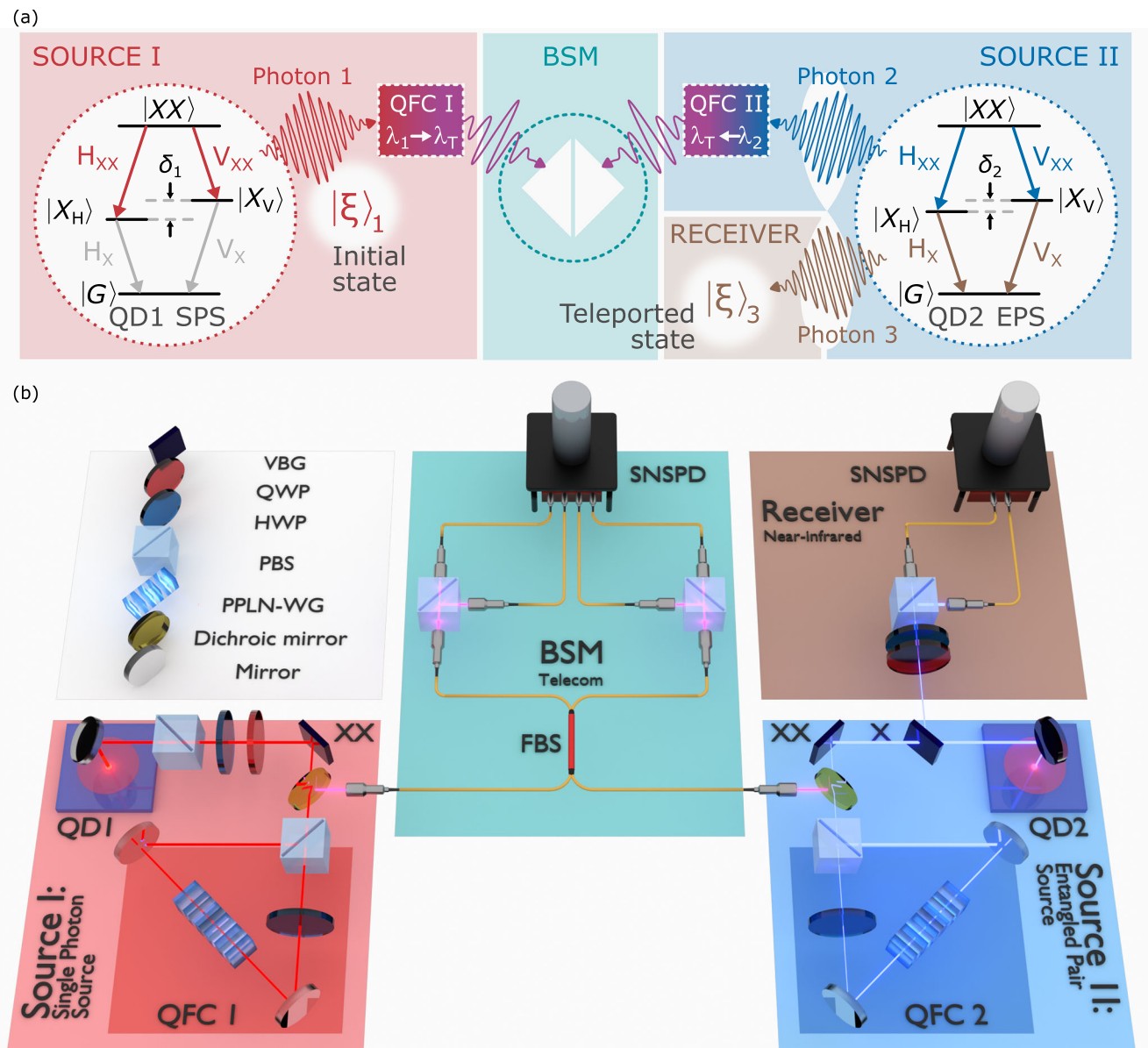

**Fig. 1 | Quantum teleportation setup. a** Schematic of the experiment where QD1 is used as a single-photon source (SPS), while QD2 is used as an entangled pair source (EPS). Two independent quantum frequency converters (QFC) are employed to convert the biexciton photons to a common telecommunication wavelength. After the Bell state measurement (BSM), the state of the single photon (named $|\xi\rangle_1$) is teleported onto the non-interfering exciton photon. **b** 3D sketch of the setup: QD1 generates a single biexciton photon, which is prepared in $|\xi\rangle_1$ using a polarizing beamsplitter (PBS) followed by a half (HWP) and quarter-wave plate (QWP), and spectrally filtered with a volume Bragg grating (VBG). The photon is frequency converted to telecommunication wavelength using polarization-preserving quantum frequency conversion, employing a periodically-poled lithium niobate waveguide (PPLN-WG), and sent to a fiber beamsplitter (FBS) for the BSM. QD2 generates an entangled photon pair: while the exciton photon is sent to the near-infrared receiver, the biexciton is frequency converted to match the wavelength of the converted biexciton photon of QD1. Polarizing beamsplitters are used in the BSM and receiver side before detection on superconducting nanowire single-photon detectors (SNSPDs).

(other combinations can be found in Supplementary Note 3). For Photon 1, 2, and 3, the single-photon count rates at the detector (summed over all measuring detectors) are $B_1 = 12.5$ kHz, $B_2 = 20.0$ kHz and $B_3 = 625$ kHz. In Supplementary Note 1I and 1J, a detailed quantification of setup and source efficiencies is provided.

**Two-photon interference after quantum frequency conversion**

Two key requirements for successful quantum teleportation are a high degree of entanglement of the EPS[45] and a high indistinguishability between the two XX photons entering the BSM. The former is intrinsic to the employed QD structure, and it is maintained by the QFC setup (entanglement fidelities up to 0.97, see Supplementary Note 1G). The

latter is mainly achieved thanks to the erasure of the initial frequency mismatch between the remote interfering XX photons via the precise spectral tuning of the pump fields in the QFC processes (being the interference visibility only eventually limited by the pumping scheme and spectral broadening).

Figure 2a depicts a high-resolution spectrum of the XX emission lines of QD1 and QD2 at telecommunication wavelength after frequency matching during QFC. A Gaussian fit function provides a linewidth of 5.2(4) GHz (4.3(1) GHz) for QD1 (QD2). The two given lines have a relative spectral offset of 0.43(27) GHz, due to pump laser drifts in the QFC setup. Decay-time measurements ($\tau_{XX}^{QD1,2} = 176$ ps, 120 ps) suggest the photon Fourier limit to be at 0.903(10) GHz and 1.322 (5)

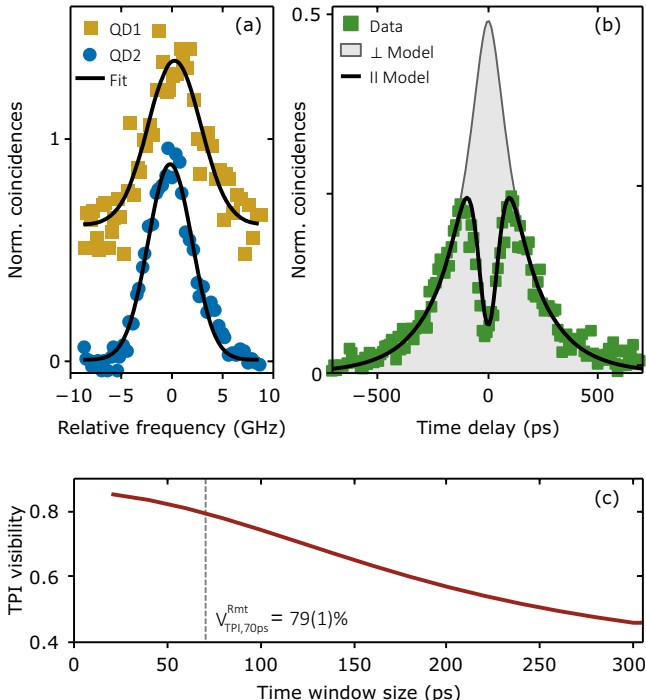

**Fig. 2 | Linewidth and interference of remote quantum light sources. a** High-resolution linewidth measurements of the QDs biexciton emissions ($|H\rangle$ component) after quantum frequency conversion, recorded with a Fabry-Pérot interferometer, also depicting their spectral overlap. The data points depict the measured results, and the solid lines are a Gaussian fit function. The data are shifted in the y-direction for clarity. **b** The center peak of a remote two-photon-interference (TPI) experiment between Photon 1 and Photon 2 with parallel polarization (green squares) is shown (see Supplementary Note 1H for full histogram). A model function as proposed by ref. 79 is used to fit these data (solid black line). The orthogonal polarization setting (gray area) is deduced from the fit results of the parallel setting. **c** Temporally post-selected remote two-photon-interference visibility $V_{\text{TPI}}^{\text{Rmt}}$ calculated from the fit results given in (**b**). The time window is centered around zero time delay.

GHz. The deviation of the measured linewidth from the Fourier limit is caused by inhomogeneous spectral broadening mechanisms[70,71] and suffices for a Gaussian approximation in the fit. The indistinguishability of the two converted emissions is probed by a two-photon-interference (TPI) experiment at the outputs of the FBS with linear polarized photons. The central peak of the correlation measurement is shown as green data points in Fig. 2b. From this, the remote TPI visibility $V_{\text{TPI}}^{\text{Rmt}}$ is evaluated (following ref. 63 and references therein). While this central peak is expected to vanish for fully indistinguishable photons, in the present case the TPI visibility is limited to 30(1)%. This has two reasons: first, the time-ordered cascade of the three-level system gives an upper limit set by the XX and X decay rates $V_{\text{TPI, max}}^{\text{Rmt}} = \gamma_{XX}/(\gamma_{XX} + \gamma_X) = 59\%$[72], and second, the inhomogeneous spectral broadening observed in the FPI measurements (the spectral broadening mechanisms are discussed in Supplementary Note 1C). The interference visibility can be increased through temporal post-selection, which can mitigate the impact of the two aforementioned mechanisms. Figure 2c depicts the interference visibility for an increasing time window centered around zero time delay. Indeed, the visibility of $V_{\text{TPI, 70 ps}}^{\text{Rmt}} = 79(1)\%$ found for a time window of 70 ps (minimal post-selection time window in the teleportation experiment discussed below) drops to 30(1)% without temporal post-selection.

## Teleportation of three conjugate polarization states
To perform the all-photonic teleportation experiment, Photon 1 is prepared in three conjugate polarization states $|\xi\rangle_1 = |H\rangle$, $|D\rangle$, and $|R\rangle$,

respectively. We measure three-photon coincidences between the BSM and Photon 3 for time windows ranging from 70 ps to 290 ps (the latter being a trade-off between covering the entire interference peak and minimizing unwanted background coincidences), resulting in averaged coincidence rates between 0.11(3) mHz and 2.5(7) mHz, respectively. The density matrix of the teleported state is reconstructed from the coincidence measurements (see methods and Supplementary Note 2C).

In Fig. 3a–c the fidelity of the teleported state $|\xi\rangle_3$ (heralded by $|\Psi^-\rangle_{1,2}$) to the three conjugate input states is calculated. Because the TPI visibility decreases as a function of integration time, as shown in Fig. 2c, the fidelity is expected to also depend on this time window. Therefore, the fidelity is evaluated for various three-photon coincidence time windows between 70 ps and 290 ps. The data points are the measurement results with error bars given by one standard deviation of a distribution obtained via a Monte-Carlo simulation (10000 runs), assuming Poissonian statistics (see Supplementary Note 2D). In an ideal scenario when teleporting $|H\rangle$ ($|D\rangle$ or $|R\rangle$) one would expect the fidelity to the $|H\rangle$ ($|D\rangle$ or $|R\rangle$) state $f^{|H\rangle \to |H\rangle}$ ($f^{|D\rangle \to |D\rangle}$ or $f^{|R\rangle \to |R\rangle}$) to be unity and the respective remaining two fidelities to be 1/2 (gray line in Fig. 3a–c). For example, a fidelity to $|H\rangle$ of 1 means the photon is maximally polarized in $|H\rangle$ and a fidelity to $|R\rangle$ of 1/2 means the photon has no polarization component in the $|R\rangle$-$|L\rangle$-basis. In Fig. 3a the fidelity of the teleported state $|H\rangle$ to state $|H\rangle$ is $f_{70\,\text{ps}}^{|H\rangle \to |H\rangle} = 0.860(23)$ for a 70 ps time window. For longer time windows, $f^{|H\rangle \to |H\rangle}$ drops only slightly but stays above 0.7. The fidelity obtained when teleporting another state, shown in Fig. 3b (Fig. 3c), is $f_{70\,\text{ps}}^{|D\rangle \to |D\rangle} = 0.630(38)$ ($f_{70\,\text{ps}}^{|R\rangle \to |R\rangle} = 0.672(34)$) and drops to 0.55 (0.6) for longer time windows. All remaining fidelities of states conjugate to the initial state of Photon 1 show deviations of $\pm 0.1$ from 1/2. The three described teleportation experiments are modeled with theoretical simulations (see methods and Supplementary Note 4 for further details). Based on this model the two main contributors to the non-unity of fidelities $f^{|H\rangle \to |H\rangle}$, $f^{|D\rangle \to |D\rangle}$ and $f^{|R\rangle \to |R\rangle}$ are the limited TPI visibility and multi-photon contributions from the QFC process (anti-Stokes Raman scattered photons generated at the target wavelength[73], see methods). For larger time windows, the TPI visibility drops (Fig. 2c) while the fraction of background counts increases, leading to the observed decrease in the fidelities. Higher teleportation fidelities of $f^{|H\rangle \to |H\rangle}$ are a result of the BSM basis choice ($|H\rangle$, $|V\rangle$) leading to additional classical correlations between the BSM and Photon 3 (see Supplementary Note 4A for a detailed explanation). Examining the corresponding density matrix $\rho$ of the teleported state is essential to understanding the variations in conjugate fidelities. A theoretical model was developed to quantify the impact of the experimental parameters on the teleportation results (more details are given in the methods section and Supplementary Note 4A). Relying on the quantum process matrix formalism for quantum teleportation with realistic QDs[45,74], this formalism provides an analytical description of the output state of the teleportation protocol, depending on the input state, and incorporates the effects of limited single-photon purity, non-zero FSS, and other decoherence processes in the QD. Here, we corrected the original output state matrix using a classical interference term that prioritizes the $|H\rangle$ state as the output from the teleportation process. The weight of this correction term is determined by the polarization mode overlap $M_p$, which accounts for the spectral and spatial distinguishability between the $|H\rangle$ and $|V\rangle$ wave packets generated by the QD due to non-zero FSS and imperfect polarization mode overlap at the FBS in the BSM. The parameters of the used QDs were either determined by spectral and radiative decay measurements (decay time of exciton $\tau_X = 171$ ps, dephasing time $T_2 = 35$ ps) or fixed based on the literature on similar QD systems (cross-dephasing time $\tau_{HV} = [1, 10]$ ns, spin scattering time $\tau_{ss} = [1, 10]$ ns)[74]. In such cases, the uncertainty

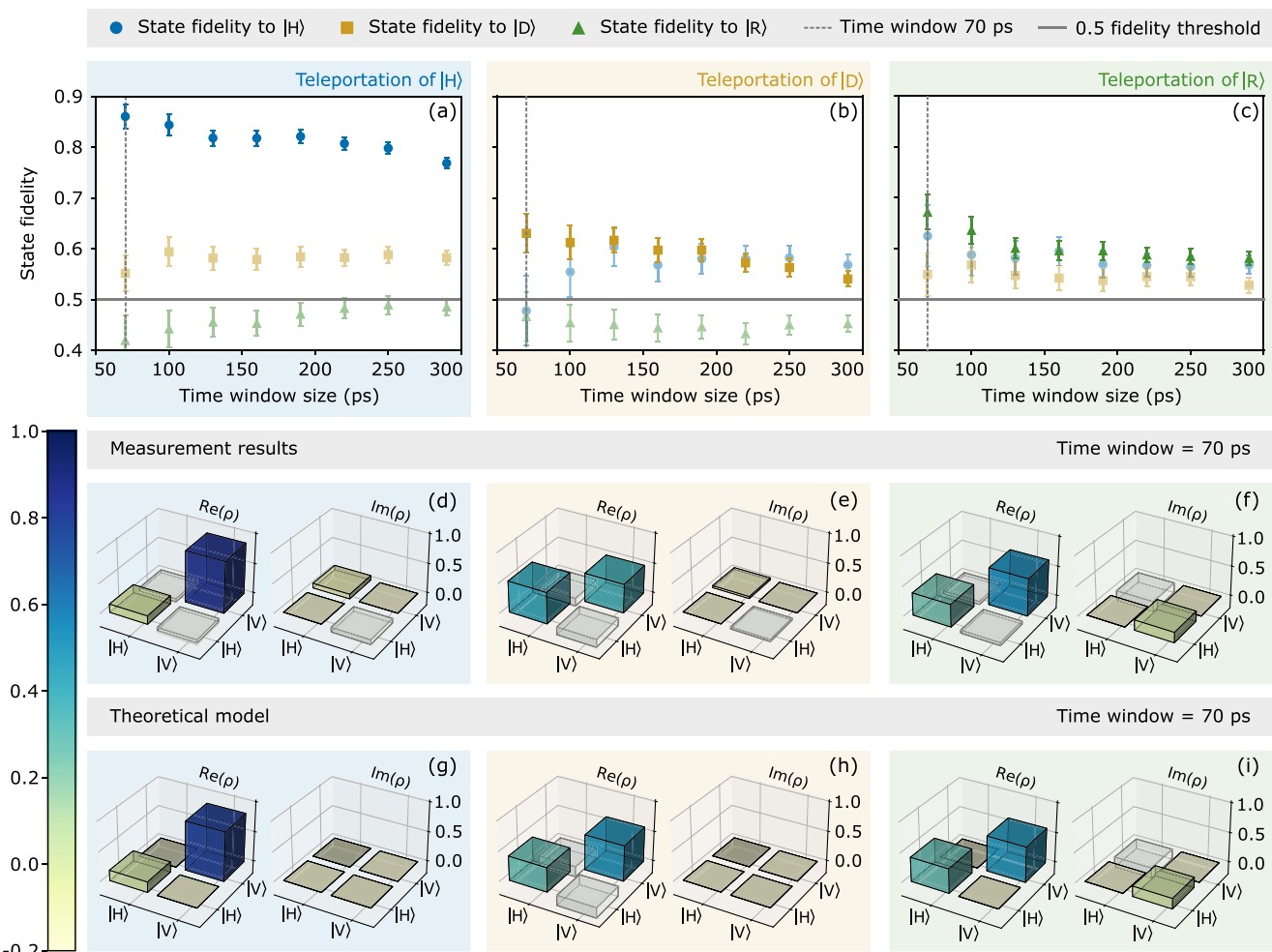

**Fig. 3 | Teleportation of three conjugate polarization states.** The teleportation experiment is repeated with three conjugate input states of Photon 1: $|\xi\rangle_1 = |H\rangle$, $|D\rangle$ and $|R\rangle$ depicted in (**a–c**). For all three experiments, the fidelity of the measured teleported state $|\xi\rangle_3$ (heralded by Bell state $|\Psi^-\rangle_{1,2}$) to the three polarization states $|H\rangle$, $|D\rangle$ and $|R\rangle$ is shown in blue, yellow and green. The error bars correspond to one standard deviation. The solid gray line at 1/2 symbolizes zero degree of polarization in the respective basis. **d–i** present real $Re(\rho)$ and imaginary parts $Im(\rho)$ of the density matrix of the teleported state $-\sigma_1\sigma_3|\xi\rangle_3$ at a 70 ps time window obtained from the measurement data (theoretical model) when teleporting $|\xi\rangle_1 = |H\rangle$, $|D\rangle$ and $|R\rangle$ respectively, before the application of any unitary transformation. To obtain the theoretical density matrices (**g–i**), the following parameters were assumed in the theory model: mode overlap $M = 0.85$, dephasing time $T_2 = 35$ ps, cross-dephasing time $\tau_{HV} = 5$ ns, spin scattering time $\tau_{ss} = 5$ ns, decay time of exciton $\tau_X = 171$ ps, TPI visibility $V = 79\%$ and ratio of true three-fold coincidences $k = 0.85$ (see methods).

interval was assumed as an input for the theoretical model, as the dephasing times $\tau_{HV}$ and $\tau_{ss}$ were not measured for the QD used. Figure 3d, e, and f (g, h and i) depict the measured (simulated) density matrices of the three teleported states, respectively for an exemplary time window of 70 ps around zero time delay, before application of any unitary transformation. The imbalance between the diagonal elements $\rho_{HH}$ and $\rho_{VV}$ in the real part of Fig. 3f, h and i arise from classical correlations, favoring the teleportation of the $|H\rangle$ state due to differences in TPI visibilities for $|H\rangle$ and $|V\rangle$ wave packets (manifesting itself in a lowered $\rho_{HH}$ and increased $\rho_{VV}$ when projecting onto $|\Psi^-\rangle_{1,2}$). This imbalance in TPI visibilities comes from the QD's non-zero FSS, non-perfect temporal wavepacket overlap, and experimental setup birefringence, which can be described by the degree of polarization mode overlap $M_p$ (see methods). We attribute the less pronounced imbalance between the diagonal elements $\rho_{HH}$ and $\rho_{VV}$ in the real part of Fig. 3e to a well-compensated experimental setup birefringence for this measurement. The measured off-diagonal elements of the real part $\rho_{HV}$ and $\rho_{VH}$ for the teleportation of $|H\rangle$ and $|R\rangle$ in Fig. 3d,f are slightly lowered compared to the theoretical model in Fig. 3g, i. For the teleportation of $|D\rangle$ non-zero off-diagonals are expected (Fig. 3e,h). The off-diagonal elements are affected by decoherence processes

within the QD, which are determined by factors such as non-zero FSS, cross-dephasing time $\tau_{HV}$, spin scattering time $\tau_{ss}$, and dephasing time $T_2$ (see Supplementary Note 4A). Additionally, the small non-zero off-diagonal elements $\rho_{HV}$ and $\rho_{VH}$ in the imaginary part for the teleported states $|H\rangle$ and $|D\rangle$ (Fig. 3d,e in comparison to Fig. 3g, h) indicate an imperfect transformation between the QD bases and the measurement bases. These effects also result in the observed deviations from 1/2 for fidelities to states conjugate to $|\xi\rangle_1$ in Fig. 3a–c. Calculated fidelities between the measured and modeled density matrices above 97% confirm the agreement between the experiment and the theoretical model (see Supplementary Note 4B).

From the results in Fig. 3a–c an average teleportation fidelity, denoted as $\bar{f} = (f^{|H\rangle \to |H\rangle} + f^{|D\rangle \to |D\rangle} + f^{|R\rangle \to |R\rangle})/3$ is determined. If one would repeat the described teleportation experiment with every possible state on the Poincaré sphere, the expected average teleportation fidelity is given by $\bar{f}$. For this reason, the average teleportation fidelity is the figure of merit for this work. The value of $\bar{f}$ indicates successful quantum teleportation when it exceeds the classical threshold of 2/3. Due to the polarization symmetry of our experimental configuration (supported by high entanglement fidelity after conversion and regular

birefringence compensation, compare Supplementary Note 1G), we expect the fidelities for the unmeasured input states ($|V\rangle$, $|A\rangle$, $|L\rangle$) to be comparable to those of the measured states, making the average over three states representative. Solid dots (the red shaded area) in Fig. 4 show the average teleportation fidelity of the experimental (modeled) teleportation process, with error bars obtained via an error propagation of standard deviations. The data correspond to the state heralded by $|\Psi^-\rangle_{1,2}$. For a time window of 70 ps, the measured average teleportation fidelity $\bar{f}_{70\text{ps}}$ equals 0.721(33), being 1.6 standard deviations above the classical threshold. The fidelity stays above this threshold up to 190 ps, which is longer than the employed XX photon decay time. For longer time windows, this value drops below the classical threshold, reaching a steady state at 0.630(12). Between 70 ps and 160 ps the modeled results closely reproduce the experimental findings. A divergence between the experiment and model, still within one standard deviation, occurs for longer time windows. Other BSM combinations can be found in Supplementary Note 3.

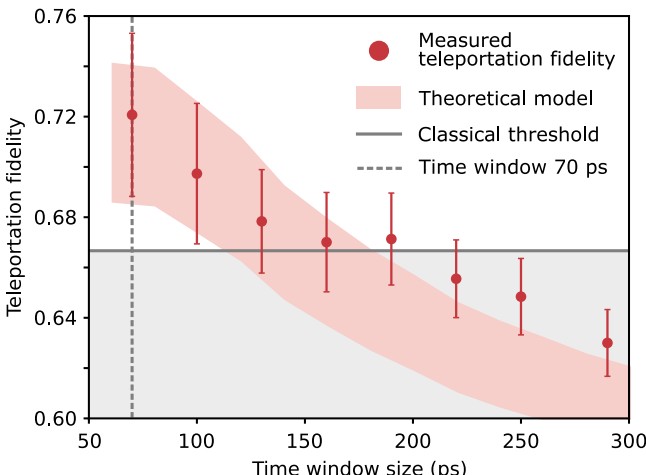

**Fig. 4 | Average teleportation fidelity.** The average fidelity over all three performed teleportation experiments $\bar{f} = (f^{|H\rangle \to |H\rangle} + f^{|D\rangle \to |D\rangle} + f^{|R\rangle \to |R\rangle})/3$, heralded by Bell state $|\Psi^-\rangle_{1,2}$ is shown. Red solid dots represent the measured results with error bars corresponding to one standard deviation. The red shaded area shows the theoretically modeled data, and the gray solid line symbolizes the classical threshold of 2/3. The uncertainty in the theoretical model arises from some parameters that cannot be precisely determined in our experimental setting. Therefore, they were based on literature[74] estimated within the following intervals to account for their variability: polarization mode overlap $M_p = [0.8, 0.9]$, cross-dephasing time $\tau_{HV} = [1, 10]$ ns, spin scattering time $\tau_{ss} = [1, 10]$ ns. The other parameters are assumed as: dephasing time $T_2 = 35$ ps and exciton decay time $\tau_X = 171$ ps.

The results achieved show successful teleportation when temporal post-selection is applied. The requirement of post-selection is mainly due to the intrinsic performance of the utilized sources and excitation method. The error of the average teleportation fidelity is mainly impacted by the counting statistics of the sources. A higher number of threefold coincidences would decrease the width of the Poissonian distributions used for the Monte-Carlo simulation, reducing the uncertainty in the average teleportation fidelity. The divergence between the experimental and theoretical results for longer time windows is primarily due to uncertainties in the spin scattering and cross-dephasing times, which could not be directly measured for the specific QD. Additional decoherence effects not captured by the simplified process matrix may also contribute, particularly at longer time scales. Thanks to the developed theory, it is possible to quantify the impact of the experimental parameters on the average teleportation fidelity, such as TPI visibility, FSS, and single-photon purity. The average teleportation fidelity achievable with optimal QD parameters is calculated to highlight the significance of these factors, see Table 1. Among them, the TPI visibility was identified as the most crucial parameter, which must be optimized for further improvement of the average teleportation fidelity. In the case of Fourier-limited sources (as shown, e.g., for gated structures[55]), where the visibility of the used QD is still limited by the decay rates of the radiative cascade to $V_{\text{TPI, max}}^{\text{Rmt}} = 59\%$ (as discussed before), the calculated fidelity surpasses the classical limit across all scenarios without requiring temporal filtering. Further improvements in TPI visibility can be achieved by incorporating a photonic structure that selectively accelerates the XX decay[75]. Assuming $\gamma_{XX} = 5\gamma_X$, which corresponds to a visibility of 83%, leads to an average teleportation fidelity of 0.73. Reducing the FSS and $g^{(2)}(0)$ further improves the achievable fidelity. The former can be minimized by mechanical strain[31] or exploiting the Stark effects[76]. The latter can be improved by reducing the conversion-related noise with narrower spectral filtering. In such a scenario (FSS = 0, $g^{(2)}(0) = 0$), an average teleportation fidelity of 0.85 could be attained. Assuming unity interference visibility, a fidelity above 0.8 is achieved in every scenario, reaching up to 0.99.

## Discussion

Here, we presented the experimental demonstration of an all-photonic quantum teleportation experiment using distinct semiconductor quantum emitters. The frequency mismatch between the remotely emitted photons is erased by employing quantum frequency conversion. Furthermore, this allows for converting the interfering photons to telecommunication wavelengths, a necessary step for upcoming long-distance implementations. These results demonstrate the maturity of quantum dot-based technology, showing an important building block for future quantum communication, i.e., the successful

**Table 1 | Theoretical average teleportation fidelity as a function of linewidth, XX and X decay rates, fine-structure splitting (FSS), and $g^{(2)}(0)$ without temporal post-selection**

| QDs parameters | $\delta_2 = 2.1\,\mu eV$, $g^{(2)}(0) \neq 0$ | $\delta_2 = 0\,\mu eV$, $g^{(2)}(0) \neq 0$ | $\delta_2 = 2.1\,\mu eV$, $g^{(2)}(0) = 0$ | $\delta_2 = 0\,\mu eV$, $g^{(2)}(0) = 0$ |
|---|---|---|---|---|
| Linewidth > FL, $\gamma_{XX}/\gamma_X = 1.4$, $V_{\text{TPI}}^{\text{Rmt}} = 30\%$ | 0.58 Present experiment without post-selection | 0.58 | 0.60 | 0.60 |
| Linewidth = FL, $\gamma_{XX}/\gamma_X = 1.4$, $V_{\text{TPI}}^{\text{Rmt}} = 59\%$ | 0.67 | 0.67 | 0.70 | 0.72 |
| Linewidth = FL, $\gamma_{XX}/\gamma_X = 5$, $V_{\text{TPI}}^{\text{Rmt}} = 83\%$ | 0.73 | 0.77 | 0.83 | 0.85 |
| $V_{\text{TPI}}^{\text{Rmt}} = 100\%$ | 0.84 | 0.87 | 0.95 | 0.99 |

First slot, current experimental parameters, with a two-photon interference (TPI) visibility of 30%, consistent with experimental results. Each row shows the impact of FSS and $g^{(2)}(0)$. TPI visibility can increase to $V_{\text{TPI, max}}^{\text{Rmt}} = 59\%$ if Fourier-limited linewidths are considered. The radiative cascade still limits this value; nevertheless, average teleportation fidelities above the classical limit can be expected. Accelerating the biexciton decay selectively (so that $\gamma_{XX}/\gamma_X = 5$) can increase the TPI visibility to 83%. The last row shows the achievable fidelities for $V_{\text{TPI, max}}^{\text{Rmt}} = 100\%$ as a function of FSS and $g^{(2)}(0)$. The following parameters were used: decay time of exciton $\tau_X = 171$ ns, polarization mode overlap $M_p = 1$; when present, dephasing time $T_2 = 35$ ns, cross-dephasing time $\tau_{HV} = 10$ ns, spin scattering time $\tau_{ss} = 10$ ns, ratio of wanted three-fold coincidences to all detected three-fold coincidence $k = 0.85$.

teleportation of a photonic state onto one photon of a polarization-entangled pair. Employing a polarization-selective BSM setup, an average teleportation fidelity of up to $\bar{f}_{70\text{ps}} = 0.721(33)$, significantly above the classical limit, is measured for a temporal post-selection window of 70 ps. The paramount factor in this realization is the capability of the QFC process to fully preserve the polarization state of the photons involved in the conversion processes. In the present experiment, fiber lengths were limited to a few meters inside the laboratory. However, due to the low loss of telecom fibers (~0.2 dB km[1]), extending the fiber length by several kilometers would introduce only minimal additional attenuation. Using an active polarization stabilization approach to compenste for stronger polarization drifts at increased fiber lengths, the teleportation fidelity values demonstrated here could be maintained. All results are fully explained by the introduced theoretical model. We foresee that a notable improvement in teleportation fidelity is within reach when employing tailored, optimized sample designs. For example, a contemporary study[77] achieved quantum teleportation with QDs into optical resonators. Although these measurements were conducted at shorter wavelengths, this approach could be used in the future, together with QFC, to further improve the signal-to-noise ratio. This will be key for the future implementation of more complex experiments, such as entanglement swapping, and the realization of more advanced quantum communication architectures.

## Methods

### Two-photon excitation and employed quantum light sources
The samples used in this work consist of droplet etching GaAs QDs[29]. Their high in-plane symmetry is reflected in low FSS. A dielectric antenna device is realized to enhance the light extraction efficiency: the semiconductor membrane embedding the QDs is glued on the bottom of a millimeter-scale GaP lens[30]. Below the membrane, a gold layer, acting as a bottom mirror, is deposited. The careful control of all layers' thicknesses allows for improved extraction and a narrow far-field emission profile (see Supplementary Note 1I). Additionally, the lens enables a tighter focusing of the excitation laser, reducing the power necessary to reach population inversion ($\pi$-pulse). Optical excitation of the biexciton state is performed via pulsed resonant two-photon excitation (TPE)[18]: this allows for the preparation of the biexciton with 45(7)% fidelity, a value estimated via the observed Rabi oscillations. More details about the excitation parameters and Rabi oscillations can be found in Supplementary Note 1A and 1D. Pulsed excitation (with a repetition rate of 304.8 MHz) results in triggered emission of entangled photon pairs via the biexciton-exciton cascade[71]. For QD1, an FSS of $\delta_1 = 10.4(2)\,\mu$eV is found, and the biexciton photon is directly projected onto the state $|\xi\rangle_1$ that will be successively teleported. The second quantum dot, QD2, is also excited under TPE, has an FSS of $\delta_2 = 2.1(3)\,\mu$eV (see Supplementary Note 1E), and the emitted photon pair has a concurrence of 0.9544(2)[18]. More information on count rates can be found in Supplementary Note 1I. It is worth mentioning that the polarization-preserving quantum frequency converters (detailed below) allow for maintaining a high degree of entanglement for QD2, as well as a well-controlled polarization of the state $|\xi\rangle_1$ of QD1 which is prepared in $|H\rangle$, $|D\rangle$ and $|R\rangle$ before conversion. Two separate closed-cycle cryostats (Cryostation s50 with Cryo-Optic configuration, from Montana Instruments) are employed to keep the samples at 6 K for the full duration of the experiments.

### Quantum Frequency Conversion
The quantum frequency converters employed in this experiment, previously used in refs. 18,65,69, utilize difference frequency generation in periodically-poled lithium niobate waveguides. In this process, QD photons are mixed with a strong pump laser to produce telecommunication wavelength photons, using a Sagnac-type setup to facilitate polarization-preserving conversion. To enable the Bell

state measurement, the photons from both QDs at 779.90(1) nm (respectively, 780.00(1) nm) are converted to a common telecommunication wavelength of 1515.53(4) nm by using separate pump lasers at 1606.74 nm (1607.1612 nm) for each conversion device. Broadband noise around the target wavelength is spectrally filtered using a bandpass filter (30 nm FWHM), and a 25 GHz VBG, resulting in a 50 kHz noise count rate for each converter. Total device efficiencies are 49% and 47%. The main noise process for the conversion scheme is anti-Stokes Raman scattering of the strong pump laser in the Lithium Niobate crystal. Due to the small spectral separation between pump and target at the low-loss telecom band around 1550 nm, the target wavelength lies within the Raman bands of the pump laser. Choosing 1607 nm as pump wavelength results in a target wavelength in the minimum of the Raman spectrum at 1515 nm. Thus, the conversion-induced noise is reduced and the signal-to-noise ratio of the experiment is improved, while the fiber losses are kept low at around 0.2 dB km[-1] compared to 0.18 dB km[-1] at 1550 nm.

### Detection System
For the described experiments, a set of six superconducting nanowire single-photon detectors is utilized (4 designed for telecom C-band operation, 2 for NIR, particularly at 780 nm). The detectors are installed in two distinct cryostats: one operates the four telecom detectors, which have 85% detection efficiency, 37 ps time resolution (full width at half maximum), and 300 Hz dark count rate. The second cryostat operates the 780 nm detectors, which have 85% detection efficiency, 44 ps time resolution, and 150 Hz dark count rate. Both detector systems are the Eos model of Single Quantum. The simultaneous operation is enabled by a Swabian Instruments time tagging unit (Time Tagger Ultra). The overall system time resolution per channel is 41 ps and 48 ps at telecommunication and NIR wavelength, respectively.

### Polarization control
The alignment of the QD polarization basis (QD2) with the measurement polarization basis is carried out by maximizing the entanglement while varying the wave plate angles of the projection units. Birefringence caused by optical fibers is compensated (between measurement runs) by sending horizontally, diagonally, or circularly polarized laser light through the setup. Monitoring the polarization at the output of the FBS in the BSM setup with a polarimeter allows for the correction of the birefringence using fiber polarization controllers and wave plates.

### Theory model
To model the quantum teleportation protocol under realistic conditions, we used a quantum process matrix formalism tailored to QD sources[45,74]. This approach yields the output density matrix $\hat{\rho}_{\text{teleported}}(|\psi_{\text{in}}\rangle)$ for a given input state $|\psi_{\text{in}}\rangle$, incorporating experimentally relevant imperfections such as limited single-photon purity, non-zero FSS, and decoherence processes. To reflect experimental alignment choices, we further corrected $\hat{\rho}_{\text{teleported}}$ with a classical interference term that enhances the fidelity for the $|H\rangle$ polarization output. The weight of this correction is given by the polarization mode overlap $M_p$, which reflects the distinguishability between $|H\rangle$ and $|V\rangle$ wave packets due to FSS, imperfect polarization overlap at the fiber beam splitter (FBS), temporal mismatches, and setup birefringence. This leads to a mixed output state upon a $|\psi^{\pm}\rangle$ Bell-state detection:

$$\hat{\rho}_{\text{out}}^{\psi^{\pm}}(|\psi_{\text{in}}\rangle) = M_p \hat{\rho}_{\text{teleported}}^{\psi^{\pm}}(|\psi_{\text{in}}\rangle) + (1 - M_p)|V\rangle\langle V|, \qquad (2)$$

This expression assumes that the TPI visibility is optimized for the $|H\rangle$ component, while the $|V\rangle$ component remains uncorrected, representing a conservative, alignment-specific scenario. The model uses both measured and literature-based QD parameters. The exciton

lifetime ($\tau_X$ = 171 ps) and pure dephasing time ($T_2$ = 35 ps) were experimentally determined, while values for the cross-dephasing time ($\tau_{HV}$ = [1, 10] ns) and spin scattering time ($\tau_{ss}$ = [1, 10] ns) were taken from similar QD systems[74], with the quoted intervals treated as uncertainty bounds in the simulation. The FSS reduces the polarization mode overlap, leading to spectral distinguishability of the $|H\rangle$ and $|V\rangle$ polarization wave packets, which was calculated as $M_p^{\mathrm{FSS}}$ = 0.94 (see Supplementary Note 4A). The model also accounted for an additional reduction in polarization mode overlap due to setup birefringence (induced by imperfect polarization control), and non-perfect temporal mode overlap due to a finite temporal resolution. Given the uncertainty in the impact of setup birefringence on $M_p$, the polarization mode overlap was assumed to be within the interval $M_p$ = [0.8, 0.9]. It is important to note that the additional $|V\rangle\langle V|$ term in the teleportation output state is specific to the chosen alignment procedure, representing a worst-case scenario. Here, the TPI visibility is maximized only for one of the polarized wave packets-in this case, the $|H\rangle$ polarized packet.

The TPI visibility was calculated for individual time window sizes from the measurement data, see Fig. 2. Similarly, the ratio of unwanted three-fold coincidences was derived by processing the measured $g^{(2)}(0)$ for all three interacting photons in individual time windows. From the dependencies of individual $g^{(2)}(0)$ on the time window size the two-photon component was estimated as[78]

$$p_2 \leq \frac{1 - Bg^{(2)}(0) - \sqrt{1 - 2Bg^{(2)}(0)}}{g^{(2)}(0)}, \qquad (3)$$

where $B$ is the QD single-photon countrate at the detector, and the assumption of $p_{\geq 3}$ = 0 is used (see Supplementary Note 4A). All possible combinations of detector clicks leading to unwanted three-fold coincidences are accounted for. The ratio of wanted coincidences to all detected coincidences is then, for a 70 ps time window, given as $k_{70\,\mathrm{ps}}$ = 0.85.

## Data availability

The data supporting the findings of this study were collected within a large corporation that is part of a broader consortium. Due to confidentiality agreements, internal data governance policies, and the large size of the dataset (approximately 10 TB), the data are not publicly available. However, they are securely stored on institutional servers and may be made available to qualified researchers upon reasonable request. Access is limited to academic use cases that align with the original research purpose and comply with consortium data-sharing policies. Requests should be directed to the corresponding author.

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

## Acknowledgements

T.S., M.V., I.N., S.K., R.J., J.H.W., C.N., P.M., and S.L.P. acknowledge the funding by the German Federal Ministry of Research, Technology, and Space (BMFTR) via Projects QR.X (16KISQ013) and QR.N (16KIS2207). Additional funding was also provided via the project EQSOTIC. This project was funded within the QuantERA II program that has received funding from the EU's H2020 research and innovation program under

the GA No 101017733, and with funding organisation BMFTR (with project number 16KIS2060K). This project has received funding from the European Union's Horizon 2020 research and innovation program under Grant Agreement no. 899814 (Qurope). T.B., M.S., and C.B. acknowledge the funding by the German Federal Ministry of Research, Technology, and Space (BMFTR) via Projects QR.X (16KISQ001K) and Q.Link.X (16KIS0864). N.J.S., W.N., G.B., and C.H. acknowledge the funding by the German Federal Ministry of Research, Technology, and Space (BMFTR) via Projects QR.X (16KISQ016). The authors gratefully acknowledge the company Single Quantum for their persistent support. We further thank Montana Instruments and Quantum Design for their support with the cryostats. We also thank Nam Tran for his contribution to creating the 3D graphics. Moreover, we would like to acknowledge Ryo Mizuta Graphics for providing resources used in the creation of the 3D graphics.

## Author contributions

T.S. and I.N. carried out the experiments with the support of S.K., R.J., and C.H. T.B. and M.S. set up the QFC devices for the experiment and supported the experimental preparations. M.V. performed the theoretical modeling. T.S., M.V., and R.J. analyzed the data. J.H.W. designed and constructed the micro-photoluminescence setup, assisted by T.S. and S.L.P. C.N. designed and built the BSM setup. N.S. grew the samples, which were processed and pre-characterized by N.S., W.N., and G.B. T.S. and S.K. pre-selected QD candidates. T.S., M.V., and S.L.P. drafted the manuscript, and all authors contributed to the final version. C.H., C.B., P.M., and S.L.P. coordinated the project.

## Funding

## Competing interests

The authors declare no competing interests.
