## [Transparent Peer Review file · Nature Communications]

Telecom-Wavelength Quantum Teleportation using Frequency-Converted Photons from Remote Quantum Dots

Corresponding Author: Mr Tim Strobel

Version 0:

Reviewer comments:

Reviewer #1

(Remarks to the Author)

The manuscript is in generally well written and reports on result interesting for the community - realization of quantum teleportation. However, some details of the experimental conditions are missing, so based on the information in the manuscript it is not possible to verify/reproduce the results in another group. Also the balance between the main text and the supplementary information should be changed towards including more information in the main text of the manuscript. In the introduction the authors should be very clear on which part is about quantum emitters in general, which about quantum dots and which about quantum dots emitting in the visible, near infrared range or at telecommunication wavelengths as it is sometimes difficult to follow. I recommend to review the reference list for the first part of the Introduction, so the first demonstrations/publications related to the topic as well as the most relevant ones (most recent progress made) are not omitted as sometimes the choice of the publications from the very broad field seems a little bit random or self-centred. The detailed comments are listed below.

The title: I would suggest to narrow the title down to specific topic of the manuscript, so use quantum dots instead of quantum emitters. If one would like to be very precise also the telecom photons from remote quantum emitters is a little bit misleading as the quantum dots do not emit telecom photons in presented work. I would consider adding down-conversion keyword in the title. This way it will be found by search engines with the specific keywords for this work. And not a general one of which this work is an example.

Introduction: it should be made clear whether the fragment achievements are for all quantum emitters, quantum dots or quantum dots in a specific spectral range. Relevance for the exact point at which there are cited should be reviewed for reference 3, 18, 44. For the defect centres in diamond works by Aharonovic seems to be missing. For quantum dot single photons sources work by Schweickert, Hanschke and Unsleber seems to be missing. In the topic of quantum dots for entangled photons work by Zeuner and Lettner seem to be missing. Ref. 30 - I would suggest to add first realization of quantum teleportation. In the topic of indistinguishable photons from quantum dots work by Thoma and Vajner are missing. In the topic of up conversion to telecom wavelengths work by Gerardot and de Greve/McMahon are missing.

Remote teleportation...

Ref 56 - it would be better to refer to first demonstration of XX-X cascade emission by Akopian. Or at least to add it.

What is the accuracy of the fine structure splitting? How such a small splitting is determined?

Why the conversion is to 1515 nm and not the maximal loss 1550 nm?

What is the efficiency of the conversion process?

Fig. 2 - Why the TPI visibility is determined via comparison between measured parallel and modelled orthogonal histogram.

These should be two independent measurements and in this case results of the first one is used to generate the second.

What are the specific inhomogeneous broadening mechanisms responsible for what is observed in experiment in this particular case (investigated QDs)? Later in the same paragraph it is stated 'mechanisms discussed before', but there is no discussion on it.

To get an idea what does the 300 ps post selection window size mean it would be important to give the repetition time of the excitation laser. Otherwise it is not known what fraction of the distance between the adjacent peaks that is. It should be stated clearly.

Why is there divergence between experimental and model results for longer time windows?

How is the polarization mode overlap determined? Why such a value of M_p , T_2 etc. has been assumed in the model?

Page 7 - it is mentioned that the theory has been developed by the Authors. If it is an important part of the work it should be described in the main text (at least assumptions and what is included and what is not included in the model).

Methods

What is the pulse duration? Pulse shape? of the excitation What laser is used? Whether any pulse shaping is applied?
Based on Methods it is not clear how M_p was determined. Should be clarified.
The information about the source of model parameters should be given directly in the main text.

Supplementary Material

1A. Spectrum of QD1 should also be shown. It would be good to show results of experiments based on which X and XX have been identified. XX binding energy should be given. Details of the excitation should be given - what power (in uW) are used? What is the excitation power density or size of the laser spot on the sample surface or optics used to focus laser on the sample surface etc. Details to reproduce the experiment. Has any off-resonant add-on to the excitation laser been used?
Fig. S1. 'unie 'a. u.' is not correct. What are the two axes in b) normalized to? The XX intensity exceeds 1? What fit function is used (black solid line)? Why are the maxima/minima for X and XX for different excitation powers?

1C. Single photon purity - it is not the purity that is measured in the experiment, but probability of multiphoton events (no vacuum fluctuations are measured)

Source brightness should be clearly defined (preferably in the main text where it was used) as different definitions are used in the community. It is important for comparison purposes.

Fig. S2a-c. What is the reason for observed short time scale bunching? Please quantify the bunching. What is the reason for large differences between peak maxima in b)?

Fig. S2d-g. The points should not be connected with unphysical broken curve. Similarly for Fig. S14.

1D. Fig. S3. What is the reason for discrepancy between $Im(q)$ before and after QFC?

Fig. S4. What is the reason for shift between maximum in orange and minimum in blue curve and vice-versa?

1F. (*) estimation - why only estimation? how were the respective values estimated?

Abovementioned points need to be satisfactorily addressed before the manuscript can be considered for publication.

Reviewer #2

(Remarks to the Author)

Strobel et al. report about experiments on all-photonic quantum teleportation using photons emitted by two spatially separated semiconductor dots (QDs). As entanglement resource for the teleportation, the biexciton-exciton radiative cascade of one QD is used, which is known to emit polarization entangled photons for small enough fine-structure splittings of the intermediate exciton level. The second QD provides single photons from its "own" biexciton state, which are interfered with biexciton photons from the second QD in a Bell-state measurement (BSM). While both QDs emit at short 780 nm, the authors apply two quantum frequency conversion (QFC) stages to convert the interfering biexciton photons to telecom C-band wavelengths, providing an intermediate solution for applications in fiber-optical networks. Using temporal post-selection, the authors are able to observe a teleportation fidelity of up to (72.1 +/- 3.3)%. The main limitation thereby being the finite two-photon interference visibility of the BSM, which, despite the frequency matching by the QFC, is limited to a raw value of 30%.

The results presented by the authors are important and timely. Previous teleportation experiments using QD sources used photons emitted from a single emitter in different emission cycles, which is much "simpler" (as higher two-photon interference visibilities can be achieved) and of limited use for practical applications. In this sense, the authors indeed make an important step in the field. On the other hand the significance of the result (in numbers) is still relatively low. The judgement on whether the work is sufficing the criteria for publication in Nat. Commun. or npj Quantum Inf. therefore is on the side of the Editor, but I believe the work should definitely be published in one of both journals. Below some comments/suggestions, the authors should address in a revised version of their manuscript.

Major Comments:

1) The extent to which the teleportation fidelity is surpassing the classical limit is unfortunately not very large, despite the rather strong temporal post-selection, and the standard deviation is substantial. What is currently limiting the measurement errors and could they be reduced to make the result more significant (I anticipate an improvement of the teleportation fidelity as not straight forward at the moment)?

2) I recommend a restructuring/revising of the abstract, which in its present form reads a bit unusual. Including more information on the actual experiment and its results will help to advertise the complex experiment, which is currently limited to 3 sentences and one number. The most part is motivation/context also with some redundancy at the beginning and the end. References are not usual for Nat Commun (but probably remanent from an earlier submission). Also I feel the strong reference to the spin-coherence results (which are doubtlessly very important) too much in this context, at least in the abstract.

3) The number of self-citations by the authors are relatively high (>10). They authors should add references in some contexts

also from other relevant groups to avoid a biased view for the reader.

Minor Comments:

- If I didn't overlook something, the authors are not yet referencing the related work by Laneve et al. [<https://arxiv.org/abs/2411.12387>], which I am sure they will cite accordingly at a later stage.
- Introduction: "Two-photon interference with light generated by distinct sources has been investigated [36–42] -> The authors are missing to cite Thoma et al. [<https://doi.org/10.1063/1.4973504>] in this context;

Reviewer #3

(Remarks to the Author)

Strobel et al. present interesting experiments that teleport the polarization state of a photon prepared from a first quantum dot, onto the polarization state of a second (entangled) photon from a second quantum dot.

In general I find the work to be well written, interesting, and of high quality.

The titular achievement of teleportation with telecom photons from remote quantum emitters is in my opinion noteworthy. There is an argument that much of what is achieved is quite close to the state of the art – for example removing the word 'telecom' from the title or replacing "quantum emitters" with "photon emitters" describes previous work cited by the authors.

However, in the context of a global quantum internet, the similar yet remote quantum emitters and telecom wavelength are an important step towards a scalable solution, and therefore have about the right impact for the field that Nature Communications demands.

Furthermore, the authors are very transparent about the context of their work, which is well referenced and highlights both previous achievements and remaining challenges. On particular challenge is of the low intensity of the teleported photons, however the efficient frequency conversion employed here suggests future use of state of the art quantum emitters which typically are sub-telecom, is highly realistic.

I find only two minor issues with the work which should be addressed, after which I recommend publication.

1. Unless I missed it, I don't see a value for the length of the fibers used in the experiment. My assumption would be a few km would be used since the additional losses this would add would be almost negligible. If only short lengths were used, then a comment that adding a few km would make little difference to the results would be helpful for some readers I would think.

2. The authors prepare H, D, and R and measure the teleportation fidelities for each. They then state that the average teleportation fidelity f_{bar} (line 53+) is the mean of these three measurements, can describe every possible state on the Poincaré sphere, and is limited to $2/3$ for the classical case. While this interpretation is prevalent in the field, this is only true for the {H,V}, {D,A}, {R,L} bases, i.e. averaged over 6 states. To illustrate, if we reduce to the {H,V,D,A} plane of the Poincaré sphere, the equivalent classical limit is $3/4$. However, if only fidelities of H, and D are measured, instead of a quantum teleporter, we could use a laser on the output, with fixed polarization between H and D (so 22.5 degrees). The intensity transmitted through the polarization analyzer would follow a \cos^2 dependence, reaching 1 midway between H and D, and ~ 0.85 at H and D $[(2+\sqrt{2})/4]$. The average exceeds 0.75 for this case – and crucially does not if A and V are added to the average. For this manuscript, I see no particular reason to expect the inverse polarization states V, A, and L to behave differently, so I suggest the authors simply clarify that assumption is required.

Version 1:

Reviewer comments:

Reviewer #1

(Remarks to the Author)

Thank you for the detailed answer and correcting the manuscript.

I have minor suggestions which would improve the manuscript further for the readers:

- Remark#9 - could you please include the informative explanation that you give in the response to the reviewers into the manuscript or SI; similarly for Remark#12 - discuss the decoherence mechanisms in your work;
- revised manuscript - page 2 - description of reference [56] - please give a value for the visibility, so the reader see directly the comparison and does not have to go to the reference to know it; "high values" is not that useful information in this context;
- page 8 - the Authors claim that the result will not be affected by adding kilometers of the fibers, but all previous study on long and especially deployed fibers show that polarization stability during propagation in the fiber becomes crucial in that case; Authors should weaken their statement and comment on this issue; it is also relevant for the discussion of (S8);
- space is missing in added Rabi oscillations (page 8);
- Supplementary material - page 3 - what is the reasoning for using \sin^2 fit instead of \sin ?

- page 4 (part E) - 'bunching is a phenomena called blinking'- something is missing in that sentence, maybe 'is caused by ...'.

After addressing these minor issues the manuscript can be published.

Reviewer #2

(Remarks to the Author)

I carefully read all three reviewer reports and the author's responses and changes made to the manuscript and the SI. My comments have been met to my full satisfaction and I believe also the other reviewer's comments have been addressed very thoroughly.

Hence I recommend publication of this work in Nature Communications.

My congratulations to the authors for this very nice work.

In response to Reviewer 1:

Reviewer 1:

The manuscript is in generally well written and reports on result interesting for the community - realization of quantum teleportation.

Our reply: We thank the Reviewer for their positive assessment of our work, and for their comments which have improved the clarity of the manuscript.

Reviewer 1 (Remark #1): However, some details of the experimental conditions are missing, so based on the information in the manuscript it is not possible to verify/reproduce the results in another group.

Our reply: Thank you for your comment. In response to this point and the more detailed suggestions you provided later in the report, we have updated the manuscript/supplement to include all relevant experimental details required for reproducibility.

Reviewer 1 (Remark #2): Also the balance between the main text and the supplementary information should be changed towards including more information in the main text of the manuscript.

Our reply: We thank the referee for her/his reply. We have included more information in the main manuscript from the supplementary material, as suggested. Changes are highlighted and discussed in more details in the following.

Reviewer 1 (Remark #3): In the introduction the authors should be very clear on which part is about quantum emitters in general, which about quantum dots and which about quantum dots emitting in the visible, near infrared range or at telecommunication wavelengths as it is sometimes difficult to follow.

Our reply: Thank you for this helpful suggestion. We have revised the introduction so that it clearly states which parts are about QDs, and the specific wavelength range of operation.

Action taken: In the second paragraph of the introduction, we labeled the type of quantum light source employed and the respective emission wavelength for each of the mentioned studies.

Reviewer 1 (Remark #4): I recommend to review the reference list for the first part of the Introduction, so the first demonstrations/publications related to the topic as well as the most relevant ones (most recent progress made) are not omitted as sometimes the choice of the publications from the very broad field seems a little bit random or self-centred. The detailed comments are listed below.

Our reply: We have modified the manuscript, adding more citations on the most recent works, as detailed below.

Action taken: We have added references Aharonovic et al., Zahidy et al., Schweickert et al., Hanschke et al., Unsleber et al., Olbrich et al., Müller et al., Zeuner et al., Lettner et al., Boschi et al., Thoma et al., Vajner et al., Morrison et al., De Greve et al., and Akopian et al. to the introduction, to ensure most relevant papers and first demonstrations are included.

Reviewer 1 (Remark #5): The title: I would suggest to narrow the title down to specific topic of the manuscript, so use quantum dots instead of quantum emitters. If one would like to be very precise also the telecom photons from remote quantum emitters is a little bit misleading as the quantum dots do not emit telecom photons in presented work. I would consider adding down-conversion keyword in the title. This way it will be found by search engines with the specific keywords for this work. And not a general one of which this work is an example.

Our reply: We thank the Reviewer for pointing out a potential ambiguity in the original title. We have revised it to “Telecom-Wavelength Quantum Teleportation using Frequency-Converted Photons from Remote Quantum Dots” to more accurately reflect the use of quantum frequency conversion.

Action taken: The title of the manuscript has been changed to “Telecom-Wavelength Quantum Teleportation using Frequency-Converted Photons from Remote Quantum Dots”.

Reviewer 1 (Remark #6): Introduction: it should be made clear whether the fragment achievements are for all quantum emitters, quantum dots or quantum dots in a specific spectral range. Relevance for the exact point at which there are cited should be reviewed for reference 3, 18, 44. For the defect centres in diamond works by Aharonovic seems to be missing. For quantum dot single photons sources work by Schweickert, Hanschke and Unsleber seems to be missing. In the topic of quantum dots for entangled photons work by Zeuner and Lettner seem to be missing. Ref. 30 - I would suggest to add first realization of quantum teleportation. In the topic of indistinguishable photons from quantum dots work by Thoma and Vajner are missing. In the topic of up conversion to telecom wavelengths work by Gerardot and de Greve/McMahon are missing.

Our reply: We thank the referee for their detailed comments on the references. At first, we targeted a general introduction encompassing works from several quantum emitters, limiting the QD-based works to the most complex experiments to avoid exceeding the suggested number of references. Although, as suggested by the referee, introduction now focuses more on QDs and gives credit to these important works, we are happy to include the references pointed out. We apologize for missing the work by Boschi et al., which has now been included.

Action taken: The mentioned works have been added to the text. Furthermore, the reference to the work of Cacciapuoti et al. has been shifted to the introduction: “A key resource in quantum communication is quantum teleportation [Bouwmeester et al., Boschi et al., Cacciapuoti et al.], [...]”. The work of Gao et al. has been shifted to the second paragraph of the introduction: “Earlier studies with single QD emitters demonstrated their potential in teleportation experiments [...] [Gao et al.]”. The work of van Loock et al. has been moved to the paragraph discussing quantum communication approaches as the work targets multiple platforms and their use in quantum repeaters: “Several material systems are currently under investigation for their role in future quantum communication [van Loock et al.]”.

Reviewer 1 (Remark #7): Remote teleportation... Ref 56 - it would be better to refer to first demonstration of XX-X cascade emission by Akopian. Or at least to add it.

Our reply: We thank the Reviewer for pointing this out and have updated the citations accordingly in the revised manuscript.

Action taken: We added the recommended citation in the first paragraph of the **Remote Teleportation with Solid-State Quantum Emitters** section: “In both cases, the QDs are excited via pulsed two-photon excitation and generate photons via the biexciton-exciton cascade ($|XX\rangle \rightarrow |X\rangle \rightarrow |G\rangle$) [Akopian et al., Young et al., Hafenbrak et al.]”.

Reviewer 1 (Remark #8): What is the accuracy of the fine structure splitting? How such a small splitting is determined?

Our reply: We thank the Reviewer for this important question. We have added a detailed description of the fine-structure splitting (FSS) measurement method to the Supplementary Information. The FSS is determined via a rotating waveplate setup, where the QD emission passes through a half-waveplate and polarizer and is recorded as a function of polarization angle. The resulting energy oscillations are fit using a sinusoidal model to extract the splitting amplitude. For QD1, we obtain $\delta_{1,X} = 10.4(2)$ μeV , and for QD2, the average value is $\delta_2 = 2.1(3)$ μeV .

Action taken: These results are now included in the **Supplementary Note 1 D**, and the uncertainties are also provided in the main manuscript.

Reviewer 1 (Remark #9): Why the conversion is to 1515 nm and not the maximal loss 1550 nm?

Our reply: The main noise process for our conversion scheme is anti-Stokes Raman scattering of the strong pump laser in the Lithium Niobate crystal. Due to the small spectral separation between pump and target for the conversion of 780 nm to the low-loss telecom band around 1550 nm, the target wavelength lies within the Raman bands of the pump laser. Choosing 1607 nm as pump wavelength results in a target wavelength in a minimum of the Raman spectrum at 1515 nm. Thus, we reduce the conversion-induced noise and improve the SBR of the experiment while keeping the fiber losses low at around 0.2 dB km^{-1} compared to 0.18 dB km^{-1} at 1550 nm.

Reviewer 1 (Remark #10): What is the efficiency of the conversion process?

Our reply: In the Methods section of the paper, we only give the external overall device efficiency (49% and 47%), as the internal process efficiency was not measured at the time of the experiment. Nevertheless, one of the QFC devices was also used and characterized in [van Leent, Tim, et al. Physical review letters 124, 010510 (2020). <https://doi.org/10.1103/PhysRevLett.124.010510>] and the second device is an exact replica of it. In the mentioned work the internal process efficiency is 96.2%.

Reviewer 1 (Remark #11): Fig. 2 - Why the TPI visibility is determined via comparison between measured parallel and modelled orthogonal histogram. These should be two independent measurements and in this case results of the first one is used to generate the second.

Our reply: Our apologies for the lack of clarity. Indeed, the TPI visibility can be evaluated by comparing the measured parallel and orthogonal cases. Otherwise, as made by several groups, the visibility can also be estimated by comparing the central coincidences around zero time delay, to the classic coincidences for larger than zero delay [Santori, C. et al. Nature 419, 594–597 (2002). <https://doi.org/10.1038/nature01086>]. As this method is accepted by the community (see also [F. Liu, et al, Nat. Nanotechnol. 13, 835 (2018). <https://doi.org/10.1038/s41565-018-0188-x>, H. Ollivier, et. al. Phys. Rev. Lett. 126, 063602 (2021). <https://doi.org/10.1103/PhysRevLett.126.063602>, N. Somaschi, et al. Nat. Photonics 10, 340 (2016). <https://doi.org/10.1038/nphoton.2016.23>]), and we validated it in previous studies even for remote sources [Weber, J.H. et al. Phys. Rev. B 97, 195414 (2018). <https://doi.org/10.1103/PhysRevB.97.195414>, Weber, J.H. et al. Nature Nanotech 14, 23–26 (2019). <https://doi.org/10.1038/s41565-018-0279-8>], we decided to adopt this method in here. Of course, as discussed in detail in the two previous studies, for remote TPI the proper normalization has to be taken into account, and even for parallel and orthogonal comparison, a fluctuation in relative intensity can become a problem. Based on our previous experience on the subject, we conducted a similar evaluation here as well.

Action taken: To avoid confusion, we added the corresponding citations where we discuss Fig. 2b: “From this, the remote two-photon-interference visibility V_{TPI}^{Rmt} is evaluated (following [Weber, et al. (2019)] and references therein).”

Reviewer 1 (Remark #12): What are the specific inhomogeneous broadening mechanisms responsible for what is observed in experiment in this particular case (investigated QDs)? Later in the same paragraph it is stated ‘mechanisms discussed before’, but there is no discussion on it.

Our reply: We thank the Reviewer for pointing this out. In these types of QD structures, the usual sources of inhomogeneous broadening are charge and spin noise [Kuhlmann, A. et al. Nature Phys 9, 570–575 (2013). <https://doi.org/10.1038/nphys2688>, Vural, H. et al. Appl. Phys. Lett. 117, 030501 (2020). <https://doi.org/10.1063/5.0010782>] as well as phonon coupling. More dedicated measurements must be performed to identify the broadening mechanisms for the presented QDs with certainty (see [T. Strobel, et al, Nano Lett. 23, 6574 (2023), <https://doi.org/10.1021/acs.nanolett.3c01658>]). In the presented case, the QDs are blinking, show bright trion lines in pure above-barrier excitation, and require a fractional amount of cw above-barrier excitation in addition to the pulsed resonant laser for efficient TPE. The latter three observations hint towards additional charge carriers being present in the vicinity of the QDs. We therefore assume that charge noise plays a role in the spectral broadening of the QD emission.

We apologize for the phrase *mechanisms discussed before* not accurately describing our intention. With this sentence, we were referring to the fact that inhomogeneous broadening is present and observed in the FPI measurements (Fig. 2a).

Action taken: The sentence in the manuscript has been changed to “[...] *the inhomogeneous spectral broadening observed in the FPI measurements.*”

Reviewer 1 (Remark #13): To get an idea what does the 300 ps post selection window size mean it would be important to give the repetition time of the excitation laser. Otherwise it is not known what fraction of the distance between the adjacent peaks that is. It should be stated clearly.

Our reply: Thank you for pointing this out. For more clarity, we have added a note when discussing the matter in the main manuscript.

Action taken: When discussing Fig. 1b we expanded the note giving the repetition rate of the experiment: “During the experimental procedure, a pulsed laser (304.8 MHz repetition rate corresponding to a 3.28 ns repetition period)[...]”.

Reviewer 1 (Remark #14): Why is there divergence between experimental and model results for longer time windows?

Our reply: We thank the referee for this insightful comment. The divergence between the experimental data and the theoretical model for longer time windows is primarily attributed to uncertainties in the spin scattering time (τ_{SS}) and the cross-dephasing time (τ_{HV}). These parameters cannot be directly measured for the specific quantum dot used in our experiment due to limitations of the experimental setup, and their detailed investigation is beyond the scope of this work.

However, based on the high uniformity of GaAs quantum dots grown via droplet epitaxy, as supported by the literature [Saimon Filipe Covre da Silva, et. al. Appl. Phys. Lett. 119, 120502 (2021).] <https://doi.org/10.1063/5.0057070>], we assumed these parameters to lie within the ranges $\tau_{SS} = [1, 10]$ ns and $\tau_{HV} = [1, 10]$ ns [D. Bauch, et. al. Advanced Quantum Technologies 7, 2300142 (2024). <https://doi.org/10.1002/qute.202300142>]. Any deviation of the actual parameters from these assumed values may significantly affect the accuracy of the theoretical model, especially at longer time windows. This is because the relevant terms in the process matrix are weighted by the k-factor and two-photon interference visibility, both of which strongly depend on the time window. For more intuition, please refer to Supplementary Equation S12.

Additionally, at longer time scales, other decoherence mechanisms may become significant. These effects are not included in our current process matrix, primarily due to a lack of detailed studies on them. Many of these mechanisms are also quantum-dot-specific.

We acknowledge the limitations of our simplified process matrix. Nonetheless, the theoretical model aligns well with the experimental results, particularly for average teleportation fidelities exceeding the classical limit. It also allows us to predict the time window at which the given experimental setup surpasses the theoretical threshold. Accurately modelling the average teleportation fidelities below the classical threshold would require a deeper understanding of the specific quantum-dot system, which is beyond the scope of this publication.

Action taken: In the discussion of Fig. 4, we added an explanation for the divergence for longer time windows: “*The divergence between the experimental and theoretical results [...]*”.

Reviewer 1 (Remark #15): How is the polarization mode overlap determined?

Our reply: Thank you for this question. We initially estimated the polarization mode overlap based on the spectral line splitting caused by the fine structure splitting (FSS). Specifically, we assumed an FSS amplitude of 2.1 μ eV and a linewidth of 4.3 GHz for the polarization-distinguished spectral lines. Using numerical integration under these assumptions, we determined the polarization mode overlap to be $M_P^{FSS} = 0.94$. This calculation is described in detail in **Supplementary Note 4 A**.

To account for potential birefringence in the experimental setup, which could introduce additional polarization mode splitting not captured by the FSS, we assumed the polarization mode overlap lies within the interval $M_P = [0.8, 0.9]$. This range accounts for the uncertainty due to possible unknown birefringence of the optical system.

Action taken: We included more information in the manuscript regarding the modeling, when discussing Fig. 3 in the main manuscript: “*A theoretical model was developed [...]*.” We also refer to the **Supplementary Note 4 A** explicitly to increase clarity.

Reviewer 1 (Remark #16): Why such a value of M_P , T_2 etc. has been assumed in the model?

Our reply: Thank you for this important question. The value of the polarization mode overlap was estimated from two main contributions: fine structure splitting (FSS) and birefringence of the optical setup. Based on measured FSS amplitude of $2.1 \mu\text{eV}$ and a spectral linewidth of 4.3 GHz , we calculated a polarization mode overlap of $M_P^{FSS} = 0.94$. To also account for potential unknown birefringence effects in the optical system, we conservatively assumed a broader range for the polarization mode overlap $M_P = [0.8, 0.9]$. For more details, please refer to our reply to **Remark #15**.

The dephasing time T_2 was derived from the measured linewidth of the exciton spectral line, $\Delta\lambda_X = 5.0 \text{ GHz}$. This linewidth consists of a natural Lorentzian component with linewidth 0.93 GHz , which corresponds Fourier-limited spectral linewidth of the measured exciton lifetime ($\tau_X = 171 \text{ ps}$) (based on previous experiments, the homogeneous broadening can be at the first approximation neglected for simplicity without drastically impacting the conclusions, see for example T. Strobel, et al, Opt. Quantum 2, 274 (2024). <https://doi.org/10.1021/acs.nanolett.3c01658>), and a Gaussian component representing the inhomogeneous broadening from dephasing processes. Through spectral deconvolution, we determined that the inhomogeneous broadening component corresponds to a linewidth of 4.50 GHz , which translates to a dephasing time of approximately 35 ps . This value was therefore used as the dephasing time T_2 in our model.

The values of spin scattering time (τ_{SS}) and the cross-dephasing time (τ_{HV}), were estimated based on the literature, as they cannot be directly measured for the specific quantum dot used in our experiment due to limitations of the experimental setup. We assumed these values to lie within the following ranges: $\tau_{SS} = [1, 10] \text{ ns}$ and $\tau_{HV} = [1, 10] \text{ ns}$ [Strobel, T., et. al. Nano Letters 23(14), 6574-6580 (2023). <https://doi.org/10.1002/qute.202300142>]. The high uniformity of GaAs quantum dots grown via droplet epitaxy is assumed [Saimon Filipe Covre da Silva et al. Appl. Phys. Lett. 119, 120502 (2021). <https://doi.org/10.1063/5.0057070>]. However, we acknowledge that these values may vary between individual quantum dots and between different sample growths. Therefore, we conservatively defined these parameters within uncertainty intervals to reflect potential sample-to-sample variation.

Action taken: We included more information in the manuscript on how these values are estimated, when discussing Fig. 3 in the main manuscript: “*A theoretical model was developed [...].*” Also, **Supplementary Note 4 A** now includes an explanation for our estimation of T_2 : “*The dephasing time ($T_2 = 35 \text{ ps}$) was derived from the measured linewidth of the exciton spectral line [...].*”

Reviewer 1 (Remark #17): Page 7 - it is mentioned that the theory has been developed by the Authors. If it is an important part of the work it should be described in the main text (at least assumptions and what is included and what is not included in the model).

Our reply: We thank the reviewer for this valuable comment. We have now included

a description of the theoretical model in the main text, highlighting the key assumptions.

Action taken: We added a description of the developed model and its main assumptions, when discussing Fig. 3 in the main manuscript: “*A theoretical model was developed [...].*”

Reviewer 1 (Remark #18): Methods What is the pulse duration? Pulse shape? of the excitation What laser is used? Whether any pulse shaping is applied?

Our reply: In accordance with the Reviewer’s comment, we have added a section in the supplementary material describing the excitation conditions during the experiment, also including details about the excitation pulse. Briefly, for pulsed two-photon excitation of both QDs, we employed a Coherent Mira Ti:sapphire laser generating 3 ps pulses. The pulses were temporally broadened to 23 ps using a 4f pulse-shaping setup, resulting in a Gaussian-like temporal and spectral profile.

Action taken: A new section in the supplement (**Supplementary Note 1 A**) called “*Excitation*” has been added, describing the excitation parameters. We explicitly refer to this new paragraph in the **Methods** section of the main text.

Reviewer 1 (Remark #19): Based on Methods it is not clear how M_p was determined. Should be clarified. The information about the source of model parameters should be given directly in the main text.

Our reply: The polarization mode overlap is estimated based on the spectral line splitting caused by the fine structure splitting (FSS). For further details, please refer to **Remark #15**.

Action taken: The utilized model parameters and their source are now given in the main text, where the theoretical model is discussed: “*The parameters of the used QDs were either determined by [...].*”

Reviewer 1 (Remark #20): Supplementary Material 1A. Spectrum of QD1 should also be shown. It would be good to show results of experiments based on which X and XX have been identified. XX binding energy should be given. Details of the excitation should be given - what power (in uW) are used?

Our reply: Spectra of QD1 in above-band excitation and the filtered XX line in two-photon excitation, together with the XX binding energies of both QDs, were added to the supplementary material. QD1 has an XX binding energy of 3.71 meV, and the value for QD2 is 3.81 meV. The identification of the X and XX lines via a fine-structure splitting measurement is now discussed in the supplementary material. The excitation

section added to the supplement also includes details about the applied excitation powers.

Action taken: We included these information in the supplementary material (see **Supplementary Note 1 B** and Fig. S1).

Reviewer 1 (Remark #21): What is the excitation power density or size of the laser spot on the sample surface or optics used to focus laser on the sample surface etc. Details to reproduce the experiment.

Our reply: We thank the Reviewer for these suggestions. Details about the experiments (including excitation powers and the objective) are given in the newly added section of the supplement. In summary, the resonant excitation laser powers were always adapted to the π -pulse, and the light was focused onto the sample with a Zeiss LD EC Epiplan-Neofluar 100x/0.75 DIC M27 objective.

Action taken: A new section (**Supplementary Note 1 A**) entitled “*Excitation*” is now written in the supplementary material.

Reviewer 1 (Remark #22): Has any off-resonant add-on to the excitation laser been used?

Our reply: Yes, an off-resonant component was added to the excitation laser. This was found to be helpful for the QD environment stabilization, resulting in a sensible increase of the count rate when in two-photon excitation.

Action taken: Details, including the corresponding powers, are now provided in the new **Supplementary Note 1 A** “*Excitation*” of the supplementary material.

Reviewer 1 (Remark #23): Fig. S1. unie 'a. u.' is not correct. What are the two axes in b) normalized to? The XX intensity exceeds 1? What fit function is used (black solid line)? Why are the maxima/minima for X and XX for different excitation powers?

Our reply: Thank you for this observation. The y-axis of the mentioned plots gives the photoluminescence intensity. It was determined by calculating the area of the respective line in the recorded spectrum. The largest measured area normalized the data for each line. The units were then changed to "arb. u." (arbitrary units). Furthermore, the unit of the x-axis was changed to the excitation power measured before the objective (excited at a rate of 76.2 MHz). This further reveals that the raw data of X and XX have their minima and maxima at the same excitation power. However, the fit to the X data appears to converge perfectly, resulting in the slightly shifted curve. The XX intensity trace appears to exceed 1 due to a vertical offset applied for visual

clarity, as mentioned in the caption. This was done to prevent overlap with the X data and does not reflect a physical value greater than 1. As this offset might lead to misinterpretations, we decided to remove it.

To fit the data, we numerically solved the optical Bloch equations, where the dephasing rate was modeled as a function of the excitation power as reported in the supplement of [Schwartz, M. et al., Nano Letters 18 (11), 6892-6897 (2018).

<https://doi.org/10.1021/acs.nanolett.8b02794>].

Action taken: We modified the figures as explained to improve the clarity. We also included in the caption the text “*The intensity was determined by calculating the area of the respective line on the spectrometer. The highest value was used as normalization factor.*”

Reviewer 1 (Remark #24): 1C. Single photon purity - it is not the purity that is measured in the experiment, but probability of multiphoton events (no vacuum fluctuations are measured)

Our reply: Thank you for pointing this out. As you correctly noted, the discussed section does not address single-photon purity. Therefore, we have renamed it to “probability of multiphoton events”.

Action taken: In **Supplementary Note 1 E** this term is now addressed as “*Probability of multiphoton events*”.

Reviewer 1 (Remark #25): Source brightness should be clearly defined (preferably in the main text where it was used) as different definitions are used in the community. It is important for comparison purposes.

Our reply: Thank you for pointing this out. Given that the setup is very complex, we have listed in the supplementary materials all individual efficiencies. Also, the source brightness is estimated, taking into account not only the extraction efficiency, but the preparation state fidelity, blinking, and excitation efficiency. This provided an overall source brightness (or source efficiency to use a broader terminology) of 1.4%. Since this value requires a precise contextualization (and can be wrongly compared to other studies), we rather keep it only in the supplementary material with all related explanations and refer to it in the main text.

Action taken: We added the measured count rates at the end of the second paragraph of the **Remote Teleportation with Solid-State Quantum Emitters** section, so that readers can directly compare to their experiments: “*For Photon 1, 2 and 3 the the single-photon count rates at the detector (summed over all measuring detectors) are $B_1 = 12.5$ kHz, $B_2 = 20.0$ kHz and $B_3 = 625$ kHz. In the supplementary material (sections 1 H and 1 I), a detailed quantification of setup and source efficiencies is provided.*” In **Supplementary Note 1 H**, we explain what source brightness is composed of:

“From these numbers, it is possible to evaluate the source brightness. As from our estimation, this also includes state preparation fidelity, excitation efficiency in presence or rate amplifier, blinking, and photonic structure extraction efficiency; we refer to it as Total Efficiency of the source. This is estimated to be around 1.4%.”

Reviewer 1 (Remark #25): Fig. S2a-c. What is the reason for observed short time scale bunching? Please quantify the bunching. What is the reason for large differences between peak maxima in b)?

Our reply: The reason for the observed bunching is blinking. Some process blocks the photoluminescence, temporarily switching off the QD emission (see for example [Santori, C. et al. Nature 419, 594–597 (2002). <https://doi.org/10.1038/nature01086>]). This process has a characteristic timescale, given by the decay constant of the observed bunching. We have now added quantitative information on the blinking behavior of QD1 and QD2 to the supplementary material. Specifically, we report that QD1 exhibits mono-exponential blinking with a decay constant of 10 ns and an optically active fraction of 0.26. QD2 shows bi-exponential blinking dynamics with decay times of 12 ns and 34 ns, resulting in an optically active fraction of 0.43. For semiconductor QDs the most common process causing blinking on this timescale is charge noise [Santori, et al., Phys. Rev. B 69, 205324 (2004). <https://doi.org/10.1103/PhysRevB.69.205324>]. During this process, surplus electrical charges can occupy the QD or get caught in traps close to the QD. This shifts the energetic structure of the QD, quenching the emission. Given that the observed blinking occurs on a timescale typical of charge noise, we attribute it to charge fluctuations in the QD environment.

Furthermore, it is important to clarify here that the reason for the large differences between peak maxima in b) is mostly due to a misalignment of the cascaded Mach-Zehnder interferometers used for rate amplification. Light passing through different optical fiber path in the interferometer causes the pulses to experience different birefringence. If the polarization of a pulse is not aligned with the QD axis the excitation will be less efficient. This effect reduces the the emission intensity of mismatched pulses creating the imbalance in the height of neighboring peaks in the cross-correlation measurement. We added an explanation clarifying this point in the supplement.

Action taken: **Supplementary Note 1 E** has been expanded, discussing the blinking and differences in peak maxima: *“The bunching of the data [...]”*.

Reviewer 1 (Remark #26): Fig. S2d-g. The points should not be connected with unphysical broken curve. Similarly for Fig. S14.

Our reply: Thank you for this valuable comment. We have updated the mentioned plots (now Fig. S4d–g and Fig. S16) accordingly by removing the unphysical connecting lines, which were only guides to the eye.

Action taken: The respective figures have been updated.

Reviewer 1 (Remark #27): 1D. Fig. S3. What is the reason for discrepancy between $\text{Im}(q)$ before and after QFC?

Our reply: We thank the Reviewer for raising this question. In the experimental setup measuring entanglement before conversion both photons were projected in a free-beam setup before passing any birefringent elements. In comparison, for the measurement, with QFC, the XX photon had to pass several meters of birefringent single-mode fiber before being projected. A polarization drift caused by birefringence of the optical fibers will lead to a polarization phase change. A phase rotation θ of one q-bit will not decrease the entanglement but will change the coherences $\rho_{14}^{\text{XX,X}}$ and $\rho_{41}^{\text{XX,X}}$ of the two-q-bit density matrix $\rho^{\text{XX,X}}$ according to:

$$\rho^{\text{XX,X}} = \frac{1}{2} \begin{pmatrix} 1 & 0 & 0 & (\cos(2\theta) - i \sin(2\theta))e^{-i \frac{\delta^2}{\hbar} t} \\ 0 & 0 & 0 & 0 \\ 0 & 0 & 0 & 0 \\ (\cos(2\theta) + i \sin(2\theta))e^{+i \frac{\delta^2}{\hbar} t} & 0 & 0 & 1 \end{pmatrix}. \quad (1)$$

For small θ the real part of the coherence is barely changing while the imaginary part of the coherence is changing approximately linear and with opposite signs. This behavior can be observed in the mentioned supplemental figure. The acquisition time for the data after QFC given in the mentioned figure was 15.13 h without any active polarization stabilization. The birefringence of the fibers was compensated once before the data recording started but not during the measurement time. We therefore conclude that the observed discrepancy in the coherences before and after QFC is caused by a polarization drift caused by birefringence of an optical fiber in the setup. This point has now been clarified in the supplementary file.

Action taken: **Supplementary Note 1 F** has been extended explaining the difference in the two scenarios: “*In the pre-conversion measurement, [...]*”.

Reviewer 1 (Remark #28): Fig. S4. What is the reason for shift between maximum in orange and minimum in blue curve and vice-versa?

Our reply: We thank the Reviewer for this interesting observation. The observed shift can be understood if we consider that both states Phi^+ and Phi^- will evolve from the $t = 0$ (fidelity close to 1 and 0 respectively) to a mixed state, where the fidelity takes value $F_\infty = 0.25$. This evolution is controlled by the photon temporal profile as well as by the oscillations induced by the non-zero fine-structure splitting. The oscillations between $F = 0$ and $F = 1$ are not symmetric around the steady-state value and are therefore damped differently, causing the observed phase shift. If they would both evolve to a fidelity which is mid-point from the starting one, i.e. 0.5, this shift would not be there. Although unphysical, we have modified our model (which otherwise fully agrees with the data in the supplementary) with the assumption that the states evolve to $F_\infty = 0.5$ instead. From the figure below, one can see that the shift disappears (dashed lines), while it is well present for the correct physical evolution towards $F_\infty = 0.25$ (solid lines). The model function used is discussed in the supplementary information of Strobel, T. et

al. Optica Quantum 2, 274-281 (2024), <https://doi.org/10.1364/OPTICAQ.530838>]. As this point has a very technical explanation, we would propose to refrain from including it in the discussion for a broad readership.

Reviewer 1 (Remark #29): 1F. (*) estimation - why only estimation? how were the respective values estimated?

Our reply: We apologize for the lack of explanations of how the values were estimated in the table mentioned. The "4-fold mismatch" efficiency is calculated as an average between the four repetitions created by the cascaded Mach-Zehnder interferometers used for rate amplification. The transmission of the objective and cryostat window were values given by the manufacturer, and were not measured by us. The transmission of the spectral filtering unit at telecom wavelength was measured with a telecom laser. This laser was spectrally matched with the QD on a spectrometer with a resolution of 6.5 GHz. The setup contained a 25 GHz reflecting VBG which is tuned via the angle. Due to the described spectral uncertainty, the reflection of the discussed VBG could be affected when changing from the laser to the QD.

Action taken: A description has been added to the caption of the addressed table.

Reviewer 1 (Remark #30): Abovementioned points need to be satisfactory addressed before the manuscript can be considered for publication.

Our reply: We thank the Reviewer for their positive assessment of our work, and for their comments which have improved the clarity of the manuscript.

In response to Reviewer 2:

Reviewer 2: Strobel et al. report about experiments on all-photonic quantum teleportation using photons emitted by two spatially separated semiconductor dots (QDs). As entanglement resource for the teleportation, the biexciton-exciton radiative cascade of one QD is used, which is known to emit polarization entangled photons for small enough fine-structure splittings of the intermediate exciton level. The second QD provides single photons from its "own" biexciton state, which are interfered with biexciton photons from the second QD in a Bell-state measurement (BSM). While both QDs emit at short 780 nm, the authors apply two quantum frequency conversion (QFC) stages to convert the interfering biexciton photons to telecom C-band wavelengths, providing an intermediate solution for applications in fiber-optical networks. Using temporal post-selection, the authors are able to observe a teleportation fidelity of up to $(72.1 \pm 3.3)\%$. The main limitation thereby being the finite two-photon interference visibility of the BSM, which, despite the frequency matching by the QFC, is limited to a raw value of 30%.

The results presented by the authors are important and timely. Previous teleportation experiments using QD sources used photons emitted from a single emitter in different emission cycles, which is much "simpler" (as higher two-photon interference visibilities can be achieved) and of limited use for practical applications. In this sense, the authors indeed make an important step in the field. On the other hand the significance of the result (in numbers) is still relatively low. The judgement on whether the work is sufficing the criteria for publication in Nat. Commun. or npj Quantum Inf. therefore is on the side of the Editor, but I believe the work should definitely be published in one of both journals. Below some comments/suggestions, the authors should address in a revised version of their manuscript.

Our reply: We thank the Reviewer for the thoughtful summary of our work and the positive assessment of its relevance and importance.

Reviewer 2 (Remark #1): Major Comments:

1) The extend to which the teleportation fidelity is surpassing the classical limited is unfortunately not very large, despite the rather strong temporal post-selection, and the standard deviation is substantial. What is currently limiting the measurement errors and could they be reduced to make the result more significant (I anticipate an improvement of the teleportation fidelity as not straight forward at the moment)?

Our reply: We agree with the Reviewer's observation. The improvement over the classical limit is indeed modest and hard to overcome due to the constraints discussed in Fig. 5 of the manuscript. We acknowledge that the standard deviation is relatively large. The error bar in the teleportation fidelity is the uncertainty of a fidelity distribution obtained via a Monte-Carlo simulation. The Monte-Carlo simulation is based on Poisson distributions of measured threefold coincidences. The threefold coincidence distributions have a certain width, translating into the width of the fidelity distribution. The width of the Poisson distributions is dictated by the absolute number of measured threefold coincidences; therefore, the width of the fidelity distribution is also given by the absolute

number of threefold coincidences. Consequently, we conclude that the error bar of the teleportation fidelity is mainly limited due to the number of threefold coincidences. Despite that, even with this large error bar, the results are clearly above the classical limit for more than 100 ps, proving successful teleportation. In future, the absolute number of threefold coincidences can be increased with longer integration times or, more favorably, by enhancing the single-photon source brightness (measured QD countrate on the detector).

Actions taken: We now explicitly address this point in the revised manuscript in the discussion of Fig. 4: *“The error of the average teleportation fidelity is mainly impacted by [...]”*.

Reviewer 2 (Remark #2): 2) I recommend a restructuring/revising of the abstract, which in its present form reads a bit unusual. Including more information on the actual experiment and its results will help to advertise the complex experiment, which is currently limited to 3 sentences and one number. The most part is motivation/context also with some redundancy at the beginning and the end. References are not usual for Nat Commun (but probably remanent from an earlier submission). Also I feel the strong reference to the spin-coherence results (which are doubtlessly very important) too much in this context, at least in the abstract.

Our reply: Thank you for this comment. We are happy to include the reviewers suggestions.

Actions taken: The redundant parts of the abstract have been removed, and more details about the experimental conditions have been added. As recommended, we have removed the citations from the abstract and excluded the spin coherence results.

Reviewer 2 (Remark #3): 3) The number of self-citations by the authors are relatively high (>10). They authors should add references in some contexts also from other relevant groups to avoid a biased view for the reader.

Our reply: We thank the Reviewer for this observation. We have carefully reviewed the manuscript and reduced the number of self-citations where possible. Additionally, we have added further references to relevant work from other research groups.

Actions taken: The number of self citations has been reduced and the works of Aharonovic et al., Zahidy et al., Schweickert et al., Hanschke et al., Unsleber et al., Olbrich et al., Müller et al., Zeuner et al., Lettner et al., Boschi et al., Thoma et al., Vajner et al., Morrison et al., De Greve et al., and Akopian et al. have been added.

Reviewer 2 (Remark #4): Minor Comments: - If I didn't overlook something, the authors are not yet referencing the related work by Laneve et al. [<https://arxiv.org/abs/2411.12387>], which I am sure they will cite accordingly at a later stage.

Our reply: We thank the referee for this comment. Indeed, the two manuscripts have been submitted back-to-back. We have now included an explicit reference to the work of Laneve et al. in the discussion section.

Actions taken: We reference the work as follows in the discussion: *“For example, a contemporary study [Laneve et al. (2024)] achieved quantum teleportation with QDs into optical resonators. Although these measurements were conducted at shorter wavelengths, this approach could be used in the future, together with QFC, to further improve the signal-to-noise ratio.”*

Reviewer 2 (Remark #5): - Introduction: "Two-photon interference with light generated by distinct sources has been investigated [36–42] -> The authors are missing to cite Thoma et al. [<https://doi.org/10.1063/1.4973504>] in this context;

Our reply: We are happy to include the work of Thoma et al.

Actions taken: The work of Thoma et al. has been added to the introduction.

We thank the Reviewer for their positive assessment of our work, and for their helpful comments. In the following, we address the Reviewer's concern about the novelty of our work in detail.

.....
In response to Reviewer 3:

Reviewer 3:

Strobel et al. present interesting experiments that teleport the polarization state of a photon prepared from a first quantum dot, onto the polarization state of a second (entangled) photon from a second quantum dot.

In general I find the work to be well written, interesting, and of high quality.

Our reply: We thank the Reviewer for the positive and encouraging feedback.

Reviewer 3 (Remark #1): The titular achievement of teleportation with telecom photons from remote quantum emitters is in my opinion noteworthy. There is an

argument that much of what is achieved is quite close to the state of the art – for example removing the word ‘telecom’ from the title or replacing “quantum emitters” with “photon emitters” describes previous work cited by the authors.

Our reply: We appreciate the Reviewer’s suggestion and are open to adjusting the title for improved clarity. One possible alternative could be: “Telecom-Wavelength Quantum Teleportation using Frequency-Converted Photons from Remote Quantum Dots”

Actions taken: The title has been changed to “*Telecom-Wavelength Quantum Teleportation using Frequency-Converted Photons from Remote Quantum Dots*”.

Reviewer 3 (Remark #2): However, in the context of a global quantum internet, the similar yet remote quantum emitters and telecom wavelength are an important step towards a scalable solution, and therefore have about the right impact for the field that Nature Communications demands.

Furthermore, the authors are very transparent about the context of their work, which is well referenced and highlights both previous achievements and remaining challenges. One particular challenge is of the low intensity of the teleported photons, however the efficient frequency conversion employed here suggests future use of state of the art quantum emitters which typically are sub-telecom, is highly realistic.

I find only two minor issues with the work which should be addressed, after which I recommend publication.

1. Unless I missed it, I don’t see a value for the length of the fibers used in the experiment. My assumption would be a few km would be used since the additional losses this would add would be almost negligible. If only short lengths were used, then a comment that adding a few km would make little difference to the results would be helpful for some readers I would think.

Our reply: We thank the Reviewer for pointing this out. Indeed, in our experiment, we used only short fiber connections within the laboratory, with typical lengths on the order of a few meters, with the longest fiber from the setup reaching the detectors of circa 50 meters. As correctly noted by the Reviewer, the additional attenuation introduced by several kilometers of telecom fiber would be negligible. To clarify this for the readers, we have added a sentence in the discussion section of the manuscript.

Reviewer 3 (Remark #3): 2. The authors prepare H, D, and R and measure the teleportation fidelities for each. They then state that the average teleportation fidelity f_{bar} (line 53+) is the mean of these three measurements, can describe every possible state on the Poincaré sphere, and is limited to $2/3$ for the classical case. While this interpretation is prevalent in the field, this is only true for the H,V, D,A, R,L bases, i.e. averaged over 6 states. To illustrate, if we reduce to the {H,V,D,A} plane of the Poincaré sphere, the equivalent classical limit is $3/4$.

However, if only fidelities of H, and D are measured, instead of a quantum teleporter, we could use a laser on the output, with fixed polarization between H and D (so 22.5 degrees). The intensity transmitted through the polarization analyzer would follow a \cos^2 dependence, reaching 1 midway between H and D, and ~ 0.85 at H and D $[=(2+\sqrt{2})/4]$. The average exceeds 0.75 for this case – and crucially does not if A and V are added to the average. For this manuscript, I see no particular reason to expect the inverse polarization states V, A, and L to behave differently, so I suggest the authors simply clarify that assumption is required.

Our reply: Thank you for this important remark. We agree that, strictly speaking, the classical fidelity bound of $2/3$ applies when averaging over six mutually unbiased polarization states (e.g., H, V, D, A, R, L).

In our experiment, we chose to measure three linearly independent states (H, D, R) under the assumption that our setup treats all polarizations symmetrically. This assumption is supported by the high fidelity of the entangled photon pair, as obtained from quantum state tomography, which strongly indicates that the setup treats all polarization components uniformly. Since maximally entangled Bell states are invariant under joint polarization rotations, we expect no systematic bias toward specific input states.

Additionally, before each measurement run, the birefringence of the optical fibers was compensated for bases H, D, and R individually to ensure equal performance across all bases.

Based on the symmetry of the experimental configuration explained above, we assume that the states V, A, and L perform similarly (within the error bar of the given fidelities), such that the average fidelity obtained from three input states H, D, and R is representative of the entire Poincaré sphere.

Actions taken: We have now clarified this point in the manuscript when discussing the average teleportation fidelity: “*Due to the polarization symmetry of our experimental configuration [...]*”

Further remarks: While reviewing the manuscript, we noticed that the given linewidths and lifetimes of the biexciton emissions of QD1 and QD2 inadvertently switched in the text and Figure 2a. However, this was purely a typographical error — the correct values were used in all evaluations. The results and conclusions remain unaffected.

We thank the Reviewer for their positive assessment of our work, and for their helpful comments. In the following, we address the Reviewer’s concern about the novelty of our work in detail.

.....

In response to Reviewer 1:

Reviewer 1 (Remark #1): Thank you for the detailed answer and correcting the manuscript.

I have minor suggestions which would improve the manuscript further for the readers:
- Remark#9 - could you please include the informative explanation that you give in the response to the reviewers into the manuscript or SI;

Our reply: We thank the Reviewer for the valuable suggestion. The detailed explanation has been added to the methods section.

Action taken: Following the Reviewers suggestion, the explanation has been added to the methods section, where the quantum frequency conversion process is discussed: *“The main noise process for the conversion [...]”*

Reviewer 1 (Remark #2): similarly for Remark#12 - discuss the decoherence mechanisms in your work;

Our reply: We are happy to include the discussion about the decoherence mechanisms in our supplementary file.

Action taken: Supplementary Note 1 C has been added to the supplementary file, discussing the decoherence mechanisms: *“In the presented type of QD structures, [...]”*. We further refer to this supplementary note on page 4 of the main file: *“(the spectral broadening mechanisms are discussed Supplementary Note 1 C)”*.

Reviewer 1 (Remark #3): - revised manuscript - page 2 - description of reference [56] - please give a value for the visibility, so the reader see directly the comparison and does not have to go to the reference to know it; "high values" is not that useful information in this context;

Our reply: Thank you for this valuable comment. We are happy to give the reported number.

Action taken: The interference visibility value from Zhai et al. is given in the second paragraph of the introduction: *“[...] with recently reported high values of 93.0(8)% interference visibility [Zhai et al.]”*

Reviewer 1 (Remark #4): - page 8 - the Authors claim that the result will not be affected by adding kilometers of the fibers, but all previous study on long and especially deployed fibers show that polarization stability during propagation in the fiber becomes crucial in that case; Authors should weaken their statement and comment on this issue; it is also relevant for the discussion of (S8);

Our reply: We thank the reviewer for this valuable remark. We agree with the Reviewer that increased fiber lengths are accompanied by additional polarization instabilities, requiring a stabilization routine.

Action taken: The statement is elaborated on in the discussion section: *“With an active polarization stabilization approach, compensating for stronger polarization drifts that come with increased fiber lengths, the teleportation fidelity values demonstrated here could be maintained.”*

Additionally, we emphasized this point in the supplementary document, where polarization drifts are discussed: *“When aiming for longer fiber lengths, especially in deployed experiments, the described polarization fluctuations become critical and require a more delicate compensation routine.”*

Reviewer 1 (Remark #5): - space is missing in added Rabioscillations (page 8);

Our reply: Thank you for pointing this out. We have carefully checked the location you indicated, but were unable to identify a missing space at that point. It is possible that this may be a formatting issue related to the PDF rendering. We will carefully verify that no typos are present in the revision of the proofs.

Reviewer 1 (Remark #6): - Supplementary material - page 3 - what is the reasoning for using \sin^2 fit instead of \sin ?

Our reply: Thank you for this valuable comment. With the amplitude, frequency, phase, and offset as open parameters the final shape of the resulting function as well as the peak-to-peak value of both model functions (\sin or \sin^2) are the same. In the case of a \sin the amplitude a has to be multiplied by a factor of two to obtain the FSS value $\delta_i = 2a$, where for the \sin^2 function the FSS directly corresponds to the FSS $\delta_i = a$. Nevertheless, a sinusoidal function models the quantity measured in a polarization series more accurately. We therefore have changed the model to a \sin function. For the reasons explained above the final FSS values stayed unchanged.

Action taken: The model function has been adjusted accordingly from \sin^2 to \sin in the supplementary section discussing the FSS:
“The data are fit with the following model:

$$f(\theta) = a \cdot \sin(b \cdot \theta + c) + d,$$

with amplitude a , frequency b , phase c and offset d , where the FSS corresponds to double the amplitude $\delta_i = 2a$.”

Reviewer 1 (Remark #7): - page 4 (part E) - 'bunching is a phenomena called blinking'- something is missing in that sentence, maybe 'is caused by ...'.

Our reply: We thank the reviewer for the comment and are happy to correct the sentence.

Action taken: The mentioned sentence in the supplementary has been corrected to: “The bunching of the data around zero time delay is caused by blinking, a phenomenon where the QD emission is temporarily quenched.”

Reviewer 1 (Remark #8): After addressing these minor issues the manuscript can be published.

Our reply: We thank the Reviewer again for his positive assessment of our work and hope his points were addressed satisfactory.

Reviewer 2 (Remark #1): I carefully read all three reviewer reports and the author’s responses and changes made to the manuscript and the SI. My comments have been met to my full satisfaction and I believe also the other reviewer’s comments have been addressed very thoroughly. Hence I recommend publication of this work in Nature Communications.

My congratulations to the authors for this very nice work.

Our reply: We are grateful for the positive evaluation and recommendation of the Reviewer.

We once again thank the Reviewers for their thoughtful comments and positive assessment.

.....